

# Why are fractional charges of orientifolds compatible with Dirac quantization?

**Yuji Tachikawa[1⋆] and Kazuya Yonekura[2]**

**1** Kavli Institute for the Physics and Mathematics of the Universe,
University of Tokyo, Kashiwa, Chiba 277-8583, Japan
**2** Faculty of Arts and Science, Kyushu University

⋆ yuji.tachikawa@ipmu.jp

## Abstract

Orientifold $p$-planes with $p \leq 4$ have fractional D$p$-charges, and therefore appear inconsistent with Dirac quantization with respect to D$(6-p)$-branes. We explain in detail how this issue is resolved by taking into account the anomaly of the worldvolume fermions using the $\eta$ invariants. We also point out relationships to the classification of interacting fermionic symmetry protected topological phases. In an appendix, we point out that the duality group of type IIB string theory is the pin$^+$ version of the double cover of GL$(2, \mathbb{Z})$.

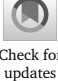
# 1 Introduction and summary

**The trouble:**  D$p$-branes and D$q$-branes are electromagnetic dual objects for $p + q = 6$, and their charges satisfy the Dirac quantization law with minimum possible value [1]. Now, it is well known that an O$p^+$-plane has the D$p$-brane charge $+2^{p-5}$, where we define its charge by integrating the flux around the angular direction $\mathbb{RP}^{8-p}$.[1] In particular, its value is fractional when $p \le 4$. Concretely, we have

$$
\begin{array}{c|ccccc}
 & \text{O4}^+ & \text{O3}^+ & \text{O2}^+ & \text{O1}^+ & \text{O0}^+ \\
\hline
\text{D}p\text{-charge} & \frac{1}{2} & \frac{1}{4} & \frac{1}{8} & \frac{1}{16} & \frac{1}{32}
\end{array}.
\tag{1.1}
$$

Naively, this contradicts the Dirac quantization law. The aim of this paper is to see how this contradiction can be resolved, by taking into account the anomaly of the worldvolume fermions of D$q$-branes. Our resolution is in part motivated by the recent development in the understanding of the classification of the interacting fermionic symmetry protected topological phases.

In this paper we only treat the O$p^+$-planes with the charge $+2^{p-5}$, and leave the discussion of the O$p^-$-planes with the charge $-2^{p-5}$ to another work.[2] Whenever we say "O$p$-plane", that means O$p^+$-plane in this paper.

**The Dirac quantization law:**  Let us first recall why the Dirac quantization law is necessary. Consider a D$q$-brane wrapped on $Y$ of spacetime dimension $q + 1$. It has a worldvolume coupling

$$
2\pi i \int_Y C_{q+1},
\tag{1.2}
$$

where $C_{q+1}$ is the Ramond-Ramond (RR) potential sourced by D$p$-branes. $C_{q+1}$ is not globally well-defined when there is a nonzero flux, so the expression (1.2) is not entirely precise.

What we actually mean is that the RR couplings when the worldvolume is $Y = Y_1$ and $Y = Y_2$ differ by the formula

$$
2\pi i \int_X F_{q+2},
\tag{1.3}
$$

where $X$ is a $(q + 2)$-dimensional manifold such that $\partial X = Y_1 \sqcup \overline{Y_2}$ and $F_{q+2}$ is the RR field strength. For illustration, see Fig. 1. There, we depicted a case when $Y_1$ is connected but $Y_2$ consists of two disconnected components. This is in general necessary, since D$q$-branes can split and merge as they interact.

The choice of such $X$ is not unique. Therefore, we need to require

$$
e^{2\pi i \int_X F_{q+2}} = e^{2\pi i \int_{X'} F_{q+2}},
\tag{1.4}
$$

for two $(q + 2)$-dimensional manifolds $X$ and $X'$ such that $\partial X = \partial X' = Y_1 \sqcup \overline{Y_2}$. Equivalently, we need to require

$$
\int_X F_{q+2} \in \mathbb{Z} \qquad \text{for any } \partial X = 0.
\tag{1.5}
$$

The condition (1.5) is not satisfied for O$p$-plane when $p \le 4$. Therefore, the phase of the partition function of D$q$-branes in this background is not well-defined, if we only consider the phase coming from the coupling to the RR flux. For the string theory to be consistent, there should be an additional source of the phase.

---

[1]For reviews on orientifolds, see e.g [2–4]. A precise formulation of orientifolds in perturbative string theory was given in [5], but in this paper we stick to a more traditional viewpoint.

[2]In the case of the O$p^-$-planes, the U(1) gauge fields on D$q$-branes also contribute to the anomaly. See [6] for the case of the O3$^-$-plane.

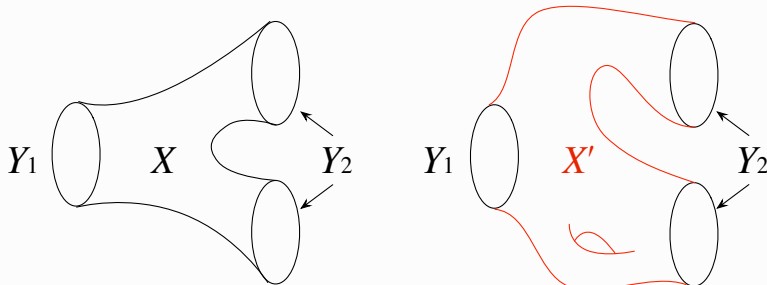

Figure 1: To compare the phases of the partition functions of D-branes wrapped around $Y_1$ and $Y_2$, we use a manifold $X$ satisfying $\partial X = Y_1 \sqcup \overline{Y_2}$. There can be multiple such choices.

**Previous analyses:** The mildest case is when $p = 4$. Then, the violation of the Dirac quantization law is only a sign, since the D4-charge of an O4-plane is 1/2. In [7,8], it was noticed that the phase of the partition function of the world-volume fermion of D2-branes in this background has a global anomaly which also produces a sign $(-1)^{\int_X w_4}$, where $w_4$ is the 4th Stiefel-Whitney class of the spacetime manifold. This means that, instead of (1.5), we need the shifted quantization condition

$$\int_X F_4 = \frac{1}{2} \int_X w_4 \mod \mathbb{Z}, \tag{1.6}$$

and this guarantees that the phase of the partition function of D2-branes are well-defined.

The main point of this paper is that this analysis can be generalized to all lower $p$ in a natural manner.[3] The essential ingredient is the realization that the most general form of the anomaly of fermions is given in terms of the $\eta$ invariant via Dai-Freed theorem [10–13],[4] and this point of view also underlies the recent development of fermionic symmetry protected topological (SPT) phases in terms of cobordisms [15].[5] Let us quickly review this framework.

**Fermion anomaly via $\eta$ invariants:** Suppose we would like to study a fermion system on $(q + 1)$-dimensional manifolds $Y$ governed by the Dirac operator $\mathcal{D}_Y$. We would like to characterize the phase difference of the partition function $Z_{\text{fermion}}(Y_1)$ and $Z_{\text{fermion}}(Y_2)$ on $Y_1$ and $Y_2$, connected by a $(q + 2)$-dimensional manifold $X$ as before: $\partial X = Y_1 \sqcup \overline{Y_2}$. We have a Dirac operator $\mathcal{D}_X$ on $X$ naturally associated to $\mathcal{D}_Y$. Then we have

$$\arg Z_{\text{fermion}}(Y_1) - \arg Z_{\text{fermion}}(Y_2) = -2\pi\eta(\mathcal{D}_X), \tag{1.7}$$

where the right hand side is the $\eta$ invariant associated to $\mathcal{D}_X$ (with a particular boundary condition at $\partial X$).

When $X$ does not have a boundary $\partial X = 0$ and $X$ itself is a boundary of a $(q + 3)$-dimensional manifold $W$, i.e. $\partial W = X$, there is also a Dirac operator $\mathcal{D}_W$ on $W$ naturally

---

[3]For a previous attempt for lower $p$, see [9].

[4]In fact, it was already pointed out in [8] that the sign $(-1)^{\int_X w_4}$ comes from the $\eta$ invariant. We should also mention that the string worldsheet has a similar issue in the Dirac quantization, where $\int_X H_3$ depends continuously on $X$ because $dH \neq 0$. Again this cancels agains the $\eta$ invariant governing the anomaly of the worldsheet fermions [14].

[5]We note that the relation between the global anomaly of fermions and the $\eta$ invariant was already used in early and important papers by Alvarez-Gaume, Della Pietra, Moore [16] and Witten [17,18]. In [18], to show the absence of the global anomaly in ten-dimensional $E_8 \times E_8$ heterotic string theory, Witten even used the vanishing of the certain cobordism group $\Omega_{11}^{\text{spin}}(K(\mathbb{Z}, 4))$, which was proved in the appendix of that paper by R. Stong.

associated to $\mathcal{D}_Y$ and $\mathcal{D}_X$, and we have the Atiyah-Patodi-Singer index theorem [19–21] (see also [22] for explanations for physicists)

$$\eta(\mathcal{D}_X) = \text{index}(\mathcal{D}_W) - \int_W P_W, \tag{1.8}$$

where index($\mathcal{D}_W$) is the index of the Dirac operator $\mathcal{D}_W$ and $P_W$ is a characteristic polynomial constructed from the curvatures on $W$.

The combination of (1.7) and (1.8) can be thought of as a refinement of the anomaly descent formalism often found in the standard quantum field theory textbooks which are available today. Namely, the equation (1.8) says that $\eta(\mathcal{D}_X)$ is more or less $-\int_X \mathcal{S}_X$ where $d\mathcal{S}_X = P_W$, and the equation (1.7) says that this Chern-Simons term controls the anomaly in the phase of the fermion partition function. The point is that the $\eta$ invariant contains additional information about the discrete constant part controlling the global anomaly, in addition to the perturbative anomaly contained in the Chern-Simons term $\int_X \mathcal{S}_X$.

**Shifted quantization law via $\eta$ invariants:**   After this quick review of the modern understanding of the fermion anomaly, let us come back to the question of the D$q$-brane again. The phase comes from the RR coupling and the fermion determinant. Therefore, the phase difference of the partition function of a D$q$-brane on $Y_1$ and $Y_2$ is given by

$$\arg Z_{\text{D}q}(Y_1) - \arg Z_{\text{D}q}(Y_2) = 2\pi \int_X F_{q+2} - 2\pi\eta(\mathcal{D}_X), \tag{1.9}$$

for an $X = Y_1 \sqcup \overline{Y_2}$. For this equation to be consistent independent of the choice of $X$, we need to have

$$\int_X F_{q+2} - \eta(\mathcal{D}_X) = 0 \quad \text{mod } \mathbb{Z}, \tag{1.10}$$

for $\partial X = 0$. The case $p = 4$ is reproduced since it is known that $\eta(\mathcal{D}_X) = \frac{1}{2}w_4 \mod \mathbb{Z}$ in this case.

Let us now come back to our original question of the orientifolds. For the consistency of the D$q$-brane in the background of an O$p$-plane, we only have to show that

$$\eta(\mathcal{D}_{\mathbb{RP}^{8-p}}) = \pm 2^{p-5} \quad \text{mod } \mathbb{Z}, \tag{1.11}$$

for the Dirac operator $\mathcal{D}_{\mathbb{RP}^{8-p}}$ appropriate for a D$q$-brane on the O$p$-plane background. The precise sign will be determined later in Sec. 2.

**The $\eta$ invariant of real projective spaces:**   These $\eta$-invariants of real projective spaces were computed by Gilkey [23–25] using the equivariant version of the Atiyah-Patodi-Singer theorem proved by Donnelly [26]. Let us sketch how the computation using the equivariant index theorem goes.[6] Suppose $X = \tilde{X}/G$ and $\tilde{X} = \partial \tilde{W}$ where $G$ acts on $\tilde{W}$ such that the fixed points of the action of $G$ are isolated points on the interior of $W$, see Fig. 2. We assume that there are no curvature contributions from smooth parts. Then the equivariant version of (1.8) says that

$$\eta(\mathcal{D}_{\tilde{X}}, g) = \text{index}(\mathcal{D}_{\tilde{W}}, g) - \sum_p P_p, \tag{1.12}$$

where $\eta(\mathcal{D}_{\tilde{X}}, g)$ and index($\mathcal{D}_{\tilde{W}}, g$) are defined by using the trace of $g \in G$ instead of the dimension of the eigenspaces of the Dirac operator. The sum $\sum_p$ is over the isolated fixed

---

[6]In Sec. 2.4 we give another argument for the computation of the relevant $\eta$ invariant without using the equivariant index theorem, following [11].

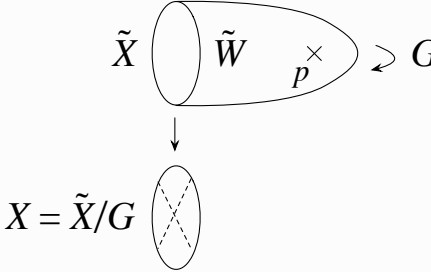

Figure 2: The $\eta$ invariant of a freely-acting quotient $X = \tilde{X}/G$ can often be computed by considering the equivariant APS index theorem on $\tilde{X} = \partial \tilde{W}$ such that the fixed points $p$ of $G$ action on $\tilde{W}$ are isolated.

points $p$ of the action of $g \in G$, and $P_p$ is the local contribution at the fixed point $p$, given solely in terms of how $g$ acts there.

In our case, we take $X = \mathbb{RP}^{q+2}$, $\tilde{X} = S^{q+2}$, and $\tilde{W} = B^{q+3}$, with $G = \mathbb{Z}_2$ acting on $\tilde{W}$ in the standard manner. The only fixed point is at the origin. We have

$$\eta(\mathcal{D}_{S^{q+2}}, g = +1) = \eta_+(\mathcal{D}_{S^{q+2}}) + \eta_-(\mathcal{D}_{S^{q+2}}), \qquad (1.13)$$

$$\eta(\mathcal{D}_{S^{q+2}}, g = -1) = \eta_+(\mathcal{D}_{S^{q+2}}) - \eta_-(\mathcal{D}_{S^{q+2}}) \qquad (1.14)$$

and

$$\eta(\mathcal{D}_{S^{q+2}}) = \eta(\mathcal{D}_{S^{q+2}}, g = +1) = 0, \qquad \eta(\mathcal{D}_{\mathbb{RP}^{q+2}}) = \eta_+(\mathcal{D}_{S^{q+2}}). \qquad (1.15)$$

Therefore we have

$$\eta(\mathcal{D}_{\mathbb{RP}^{q+2}}) = \frac{1}{2} P_{p=0}. \qquad (1.16)$$

What remains is the determination of the local contribution $P_{p=0}$ at the origin $p = 0$. This can be done by a careful computation of $P_{p=0}$ since a general formula for it is known. But it can be done more easily by a trick. We consider $\tilde{W} = T^{q+3}$ with the standard action of $\mathbb{Z}_2$ flipping every direction. We have

$$0 = \text{index}(\mathcal{D}_{T^{q+3}}, g = -1) - 2^{q+3} P_0. \qquad (1.17)$$

Now, the fermions on a D$q$-brane arise from the dimensional reduction (or T-dual) of a Majorana-Weyl fermion from ten dimensions. As such, the number of components of the fermions is real 16 or complex 8 in $q + 1$ dimensions. These 8 components are for chiral fermions, and hence the corresponding Dirac (instead of Weyl) fermion in $q + 1$ dimensions has $2 \times 8 = 16$ components. Then the Dirac fermion in two dimensions higher (i.e. $q + 3$ dimensions) has $2 \times 16 = 32$ components. It turns out (see Sec. 2) that the $\mathbb{Z}_2$ action on the fermion components is given by the chirality operator $\bar{\Gamma}_{T^{q+3}}$ on $T^{q+3}$ up to a sign as $g : \Psi(\vec{z}) \mapsto \pm \bar{\Gamma}_{T^{q+3}} \Psi(-\vec{z})$. The chirality operator $\bar{\Gamma}_{T^{q+3}}$ in this transformation is required by the fact that the Dirac operator $\mathcal{D}_{T^{q+3}} = i\Gamma^I \partial_I$ on $T^{q+3}$ is equivariant under $g$ as $\mathcal{D}_{T^{q+3}} \circ g = g \circ \mathcal{D}_{T^{q+3}}$. Therefore, $\text{index}(\mathcal{D}_{T^{q+3}}, g = -1) = \pm 32 = \pm 2^5$, and $P_0 = \pm 2^{2-q} = \pm 2^{p-4}$. Plugging this into (1.16), we indeed get (1.11).

That said, we have been cavalier in the various signs and in particular the action of $\mathbb{Z}_2$ on the fermions during the rough outline we gave above. We will spend Sec. 2 to carefully fix the conventions and the signs.

**Organization of the paper:** The remainder of our paper consists of two more sections and an appendix, most of which can be read independently:

- In Section 2, we carefully specify our conventions concerning index theorems and string theory, determine the $\mathbb{Z}_2$ action relevant for the O$p$-plane, and carry out the strategy outlined above. We confirm that all the fermions contribute to the anomaly by the same sign, and we get $2^{p-5} - 2^{p-5} = 0$ instead of $2^{p-5} + 2^{p-5} \neq 0$. This will be done in a uniform computation independent of the dimensionality $p$.

- In Section 3, we interpret the global fermion anomaly we detected using the $\eta$ invariants in terms of cobordism invariants classifying the interacting fermionic SPT phases. We see that it illustrates the dimensional hierarchy of SPTs with (anti)unitary $\mathbb{Z}_2$ symmetries squaring to $(\pm 1)^F$ discussed in Sec. 8 of [15].

- The appendix A, we discuss the precise nature of the symmetry group of the type IIB string theory, which is usually called the SL$(2, \mathbb{Z})$. We argue that it is a pin$^+$ version of the double cover of GL$(2, \mathbb{Z})$.

## 2 Careful analysis of anomalies

### 2.1 Conventions concerning index theorems

First let us fix the conventions related to index theorems. We follow the conventions in [12]; see in particular the Appendix A and B of that paper. We work entirely in Euclidean signature. The spacetime dimension is denoted by $d$. We denote the spacetime indices as $I, J, M, \cdots$. These run over $0, 1, \cdots, d-1$. We use $I = 0$ even in Euclidean signature.

We take the gamma matrices $\Gamma^I$ to be hermitian following the standard physics convention. Also, we denote the chirality operator as $\overline{\Gamma}$ which gives the $\mathbb{Z}_2$-grading relevant for the index problem; This is the one which is usually denoted as $\gamma_5$ in four dimensions. It must satisfy $\overline{\Gamma}^2 = 1$ and $\{\overline{\Gamma}, \Gamma^I\} = 0$. We call $\overline{\Gamma} = +1$ as positive chirality and $\overline{\Gamma} = -1$ as negative chirality. We take the Dirac operator to be

$$\mathcal{D} = i\Gamma^I D_I, \tag{2.1}$$

where $D_I$ is the covariant derivative.

On non-oriented manifolds, we need an additional gamma matrix which we denote by $\widetilde{\Gamma}$ because of the following reason. On a flat space, consider the reflection $x^J \to -x^J$ for some $J$. We need a corresponding operation in the Pin$^\pm$ groups acting on a fermion $\Psi$. If we denote this action as $\Psi(\cdots, x^J, \cdots) \mapsto E_J \Psi(\cdots, -x^J, \cdots)$, then the matrix $E_J$ must satisfy (i) $\overline{\Gamma} E_J = E_J \overline{\Gamma}$ so that the $\mathbb{Z}_2$-grading is well-defined, and also satisfy (ii) $\Gamma^I E_J = E_J \Gamma^I$ ($I \neq J$) and $\Gamma^J E_J = -E_J \Gamma^J$ for the Dirac operator to behave covariantly. These equations are satisfied if we define $E_J$ as $E_J = \alpha \widetilde{\Gamma} \Gamma^J$, where $\alpha = \pm i$ for Pin$^+$ and $\alpha = \pm 1$ for Pin$^-$. It is possible to realize Pin$^\pm(d)$ as a subgroup of Spin$(D)$ for some $D > d$, and then $\widetilde{\Gamma}$ is given by the additional gamma matrices in Spin$(D)$.

**APS index theorem:** The index theorem is given as follows. In this paper we only need the index theorems on even dimensional oriented manifolds or odd dimensional non-oriented manifolds, so we restrict our attention to these cases. If we denote the number of zero modes with positive chirality ($\overline{\Gamma} = +1$) and negative chirality ($\overline{\Gamma} = -1$) by $n_+$ and $n_-$ respectively, we have the APS index theorem of a Dirac operator $\mathcal{D}_W$ on a manifold $W$ with boundary $\partial W = X$

as

$$\text{index}(\mathcal{D}_W) = n_+ - n_- = \int_W \hat{A}(R)\text{ch}(F) + \eta(\mathcal{D}_X), \tag{2.2}$$

where $\text{ch}(F) = \text{tr}\exp(\frac{i}{2\pi}F)$ is the Chern character in the standard convention, $\hat{A}(R)$ is the standard Dirac genus, and $\mathcal{D}_X$ is the boundary Dirac operator. The $\eta$ invariant of a self-adjoint operator $\mathcal{D}$ with the set of discrete spectrum of eigenvalues $\text{Spec}(\mathcal{D})$ is defined as

$$\eta(\mathcal{D}) = \frac{1}{2}\left(\sum_{\lambda \in \text{Spec}(\mathcal{D}),\ \lambda \neq 0} \text{sign}(\lambda) + \dim(\text{Ker}\,\mathcal{D})\right), \tag{2.3}$$

with an appropriate regularization for the sum over all eigenvalues $\lambda \in \text{Spec}(\mathcal{D})$. However, the equation (2.2) needs more explanations about the orientation and the precise definition of $\mathcal{D}_X$.

In even dimensions $d = 2n$, we define the totally anti-symmetric tensor by

$$\epsilon^{I_1 I_2 \cdots I_{2n}} = \frac{1}{i^n 2^n} \text{tr}\left(\overline{\Gamma}\Gamma^{I_1}\Gamma^{I_2}\cdots\Gamma^{I_{2n}}\right). \tag{2.4}$$

Then this totally anti-symmetric tensor $\epsilon^{I_1 \cdots I_d}$ sets the orientation of the manifold. In the first term of (2.2), we have to use this orientation. This can be seen by e.g. carefully deriving the index theorem by the standard method of Fujikawa, or by considering $W = \Sigma \times \cdots \times \Sigma$ where $\Sigma$ is a 2d Riemann surface and using the Riemann-Roch theorem. In odd dimensions, $\hat{A}(R)\text{ch}(F)|_{d=\text{odd}}$ is zero and hence the orientation does not matter.

The boundary Dirac operator $\mathcal{D}_X$ is related to the bulk Dirac operator $\mathcal{D}_W = i\Gamma_W^I D_I$ as follows. Let $x$ be a coordinate such that the boundary $X$ is located at $x = 0$, and the interior of $W$ is $x < 0$. The metric near the boundary is assumed to be of the form $ds_W^2 = dx^2 + ds_X^2$. Then by using the data near the boundary, we define

$$\mathcal{D}_W = i\Gamma_W^x(\partial_x + \widehat{\mathcal{D}}_X), \qquad \widehat{\mathcal{D}}_X = \Gamma_W^x \Gamma_W^{I'} D_{I'}, \qquad \mathcal{D}_X = \widehat{\mathcal{D}}_X P_W^+ = i\Gamma_X^{I'} D_{I'}, \tag{2.5}$$

where we used

$$P_W^\pm = \frac{1}{2}(1 \pm \overline{\Gamma}_W), \qquad \Gamma_X^{I'} = -i\Gamma_W^x \Gamma_W^{I'} P_W^+. \tag{2.6}$$

The index $I'$ runs over the subset of $\{0, 1, \cdots, d-1\}$ which are orthogonal to the direction specified by $x$. Notice that $P_W^+$ commutes with $\widehat{\mathcal{D}}_X$, and hence the eigenmodes of $\widehat{\mathcal{D}}_X$ are preserved under the projection by $P_W^+$. Thus $\mathcal{D}_X$ is a self-adjoint operator.

Notice the following point. On non-oriented manifolds, there was no orientation and hence there was no definite way to pick the sign of $\overline{\Gamma}_W$. If we change $\overline{\Gamma}_W \to -\overline{\Gamma}_W$, the definition of positive and negative chirality is exchanged and hence the index becomes $\text{index}(\mathcal{D}_W) \to -\text{index}(\mathcal{D}_W)$. Let us check how this is consistent with the index theorem. For simplicity we assume that $\text{Ker}\,\mathcal{D}_X$ is empty, although more general case can also be analyzed. If we change $\overline{\Gamma}_W \to -\overline{\Gamma}_W$, the projection operator changes as $P_W^+ \to P_W^-$. Now, the vector spaces with $P_W^+ = 1$ and $P_W^- = 1$ can be exchanged by the multiplication by $\Gamma_W^x$. This multiplication by $\Gamma_W^x$ also changes the boundary Dirac operator as $\widehat{\mathcal{D}}_X \to \Gamma_W^x \widehat{\mathcal{D}}_X \Gamma_W^x = -\widehat{\mathcal{D}}_X$, and hence the $\eta$ invariant is changed as $\eta(\mathcal{D}_X) \to \eta(-\mathcal{D}_X) = -\eta(\mathcal{D}_X)$ where we used $\text{Ker}\,\mathcal{D}_X = \varnothing$. Therefore the index formula (2.2) behaves nicely under this change. This gives a quick motivation for the insertion of the projector $P_W^+$ in the definition of $\mathcal{D}_X$. Without this projector, we have $\eta(\widehat{\mathcal{D}}_X) = 0$.

The coefficient of the term $\eta(\mathcal{D}_X)$ in (2.2) in the present convention can be checked e.g. in a simple example of a 2d fermion coupled to a U(1) background field, by taking $W = D^2$,

$X = S^1$, and $\mathcal{D}_W = i(\sigma^1 D_1 + \sigma^2 D_2)$. In this case, one can compute the eta invariant directly from the definition, and check that the right-hand-side of (2.2) takes values in integers $\mathbb{Z}$ for any background U(1) field. See also Appendix B of [12] for more discussions on general oriented case.

For the APS index theorem to hold, we have to impose the APS boundary condition. For the eigenmodes $\Psi$ of $\mathcal{D}_W$, we require the boundary condition

$$\Psi|_X \in \left\{ \sum_{\lambda \geq 0} c_\lambda P_W^+ \Psi_\lambda^X + \sum_{\lambda > 0} c'_\lambda P_W^- \Psi_\lambda^X \right\}, \tag{2.7}$$

where $\Psi_\lambda^X$ are the eigenmodes of $\widehat{\mathcal{D}}_X$, i.e. $\widehat{\mathcal{D}}_X \Psi_\lambda^X = \lambda \Psi_\lambda^X$. Namely, the restriction of $\Psi$ on the boundary $X$ must be spanned by eigenmodes of non-negative eigenvalues for $\overline{\Gamma}_W = +1$ and positive eigenvalues for $\overline{\Gamma}_W = -1$. This is imposed so that if the manifold $W$ is extended by a cylinder region $[0, \infty) \times X$, then the zero modes (which are solutions of $(\partial_x + \widehat{\mathcal{D}}_X)\Psi = 0$ in the cylinder region) do not grow at infinity.

From this boundary condition, we can give a crude idea of why the boundary term $\eta(\mathcal{D}_X)$ appears in the APS index theorem (2.2). Consider the heat kernel method for the index computation,

$$\text{index}(\mathcal{D}_W) = \lim_{t \to +0} \text{Tr}(\overline{\Gamma}_W e^{-t(\mathcal{D}_W)^2}). \tag{2.8}$$

Near the boundary, the relevant modes are given by (2.7). The modes with $\overline{\Gamma}_W = +1$ and $\lambda \geq 0$ contribute positively for this heat equation, while the modes with $\overline{\Gamma}_W = -1$ and $\lambda > 0$ contributes negatively. However, the modes with $\overline{\Gamma}_W = -1$ and $\lambda > 0$ can be mapped to the modes with $\overline{\Gamma}_W = +1$ and $\lambda < 0$ by using $\Gamma_W^x$ as mentioned above. Therefore, if we map all the modes to $\overline{\Gamma}_W = +1$, the contribution of the mode with an eigenvalue $\lambda$ of $\mathcal{D}_X$ to the heat equation is proportional to $\text{sign}(\lambda)$ with a positive proportionality constant. (Here we define $\text{sign}(\lambda = 0) = +1$.) Their sum is the $\eta$ invariant $\eta(\mathcal{D}_X)$ up to a positive proportionality constant. It requires more work to show that the positive proportionality constant is just as in (2.2). See also [22] for more physical interpretation of the boundary $\eta$ invariant.

**Dai-Freed theorem:** Now consider a manifold $X$ with boundary $\partial X = Y$. We assume that there is a coordinate $y$ such that $Y$ is located at $y = 0$ and the interior of $X$ is in the region $y < 0$. The metric is assumed to be of the form $ds_X^2 = dy^2 + ds_Y^2$

If we are given a Dirac operator $\mathcal{D}_X = i\Gamma_X^{I'} D_{I'}$ (which need not be $\mathbb{Z}_2$-graded), we define the boundary Dirac operator $\mathcal{D}_Y$ as

$$\mathcal{D}_X = i\Gamma_X^y(\partial_y + \mathcal{D}_Y), \qquad \mathcal{D}_Y = \Gamma_X^y \Gamma_X^{I''} D_{I''} = i\Gamma_Y^{I''} D_{I''}, \tag{2.9}$$

where

$$\Gamma_Y^{I''} = -i\Gamma_X^y \Gamma_X^{I''}. \tag{2.10}$$

We also define the boundary chirality operator $\overline{\Gamma}_Y$ and the projection operators $P_Y^\pm$ as

$$\overline{\Gamma}_Y = \Gamma_X^y, \qquad P_Y^\pm = \frac{1}{2}(1 \pm \overline{\Gamma}_Y). \tag{2.11}$$

In the above setup, the Dai-Freed theorem states the following. Let

$$\mathcal{D}_Y^{-+} = P_Y^- \mathcal{D}_Y P_Y^+ \tag{2.12}$$

be a chiral Dirac operator. Then the combination

$$\det(\mathcal{D}_Y^{-+})\exp(-2\pi i\eta(\mathcal{D}_X)) \tag{2.13}$$

takes values in $\mathbb{C}$, although each of $\exp(-2\pi i\eta(\mathcal{D}_X))$ and $\det(\mathcal{D}_Y^{-+})$ takes values in one-dimensional vector spaces called the determinant line and its inverse.

More physically, this means the following. The chiral Dirac operator $\mathcal{D}_Y^{-+}$ may suffer from anomalies, and $\det(\mathcal{D}_Y^{-+})$ may not be well-defined as a number in $\mathbb{C}$. However, if we extend $Y$ to a manifold $X$ such that $\partial X = Y$, then we can make it well-defined by multiplying it by the exponentiated $\eta$ invariant $\exp(-2\pi i\eta(\mathcal{D}_X))$ in the bulk. The $\eta$ invariant itself suffers from some ambiguity on manifolds with boundary, since there is no preferred choice of boundary conditions. The ambiguities of $\det(\mathcal{D}_Y^{-+})$ and $\exp(-2\pi i\eta(\mathcal{D}_X))$ cancel each other.

The Dai-Freed theorem has a very nice physical interpretation [11]. Consider a massive fermion $\Psi$ whose Euclidean action is given by

$$-S_E = -\int \bar{\Psi}(-i\mathcal{D}_X + M)\Psi. \tag{2.14}$$

If we take the Pauli-Villars regulator mass as positive, $M = +\Lambda$ and $\Lambda \to +\infty$, and the physical fermion mass as negative and very large $M = -\Lambda$, the partition function is given by [16]

$$Z_{\text{bulk}} = \frac{\det(-i\mathcal{D}_X - \Lambda)}{\det(-i\mathcal{D}_X + \Lambda)} = \prod_{\lambda\in\text{Spec}(\mathcal{D}_X)} \frac{(-i\lambda - \Lambda)}{(-i\lambda + \Lambda)} = e^{-2\pi i\eta(\mathcal{D}_X)}, \tag{2.15}$$

where we took the regularized version of $\text{sign}(\lambda)$ appearing in the definition of the eta invariant as

$$\text{sign}(\lambda)_{\text{reg}} = \frac{1}{\pi}\arg\left(-\frac{(-i\lambda + \Lambda)^2}{\lambda^2 + \Lambda^2}\right), \qquad -1 < \text{sign}(\lambda)_{\text{reg}} \le 1, \tag{2.16}$$

which gives the usual sign $\lambda/|\lambda|$ for $\lambda \ll \Lambda$ and which goes to zero, $\text{sign}(\lambda)_{\text{reg}} \to 0$ for $|\lambda| \gg \Lambda$. Therefore, the partition function of the above massive fermion is given by $Z_{\text{bulk}} = e^{-2\pi i\eta(\mathcal{D}_X)}$.

Now we take the fermion mass $M$ to depend on the position as follows. Let $\epsilon > 0$ be a very small length scale. We take $M(y)$ to be $M(y) = -\Lambda$ for $y < -\epsilon$, and $M(y) = +\Lambda$ for $y > +\epsilon$. Namely, inside $X$ the mass is negative as above, but outside $X$ (i.e. $y > 0$) it is positive, and $M(x)$ changes rapidly near the boundary $Y$. In this case, there is a localized chiral fermion on $Y$ which is given by the solutions of

$$(\Gamma_X^y\partial_y + M(x))\Psi = 0, \qquad \Psi \propto \exp\left(-\Gamma_X^y\int_0^y M(y')dy'\right). \tag{2.17}$$

For the solutions to be localized near $Y$, they must satisfy $\Gamma_X^y\Psi = +\Psi$, and hence they have the positive chirality under $\overline{\Gamma}_Y = \Gamma_X^y$.

The total system is as follows. In the bulk $x < 0$, we have the partition function $Z_{\text{bulk}} = e^{-2\pi i\eta(\mathcal{D}_X)}$. On the boundary, the chiral fermion with positive chirality $\overline{\Gamma}_Y = +1$ gives the partition function $Z_{\text{boundary}} = \det(\mathcal{D}_Y^{-+})$. The region $x > 0$ outside of $X$ does not contribute because of the cancellation between the physical fermion and the Pauli-Villars field. Therefore, the total partition function of the system is

$$Z = Z_{\text{boundary}}Z_{\text{bulk}} = \det(\mathcal{D}_Y^{-+})\exp(-2\pi i\eta(\mathcal{D}_X)). \tag{2.18}$$

This should be well-defined since the total system is a non-chiral theory and has a well-defined UV regularization. This is one aspect of the physics of the Dai-Freed theorem [11]. See also [12] for another aspect.

## 2.2 Conventions concerning string theory

**Chirality in type IIB:** In type IIB superstring theory, we require that the supercharges $Q, \tilde{Q}$ have the chirality $\overline{\Gamma} = -1$,

$$\overline{\Gamma}Q = -Q. \tag{2.19}$$

This is just a choice. Then, all other chiralities are automatically fixed.

For our purposes, what are important are the gaugino on D-branes and the duality equations for the field strengths of RR fields $C$. First, the chirality of gaugino is determined as follows. Let us consider type I superstring theory. Its supercharge $Q^{(\text{type I})}(= Q + \tilde{Q})$ has the same chirality as the one in type IIB. Moreover, the gaugino $\lambda$ of the SO(32) vector multiplet and the gauge field $A_I$ are roughly related as

$$
\begin{aligned}
&[Q^{(\text{type I})}, A_I] \sim \Gamma_I \lambda, \\
&\{\lambda, (Q^{(\text{type I})})^T \mathsf{C}\} \sim (\partial_I A_J - \partial_J A_I)\Gamma^I \Gamma^J,
\end{aligned} \tag{2.20}
$$

where $\mathsf{C}$ is a charge conjugation matrix whose defining property is that $\mathsf{C}\Gamma_I \mathsf{C}^{-1} = -(\Gamma_I)^T$, $\mathsf{C}\overline{\Gamma}\mathsf{C}^{-1} = -\overline{\Gamma}$, and $(\Psi^T \mathsf{C})\Gamma^I \Psi'$ transforms as a Lorentz vector for two spinors $\Psi$ and $\Psi'$ of the same chirality. This fixes the chirality of the gaugino of the "D9"-brane as

$$\overline{\Gamma}\lambda = +\lambda. \tag{2.21}$$

The fact that the gauginos have chirality $\overline{\Gamma} = +1$ is valid for any D$p$-brane under the following interpretation. The gamma matrices $\Gamma^0, \cdots, \Gamma^p$ are for the tangent bundle of the worldvolume, while $\Gamma^{p+1}, \cdots, \Gamma^9$ are for the normal bundle or equivalently the R-symmetry. One can simply repeat the above argument, with the understanding that $A_I$ for $I$ being normal directions are scalars. For the transformation (2.20) to be valid, we use the convention of T-duality which will be discussed later.

Next, we determine the chirality of the 5-form field strength $F_5 = dC_4$. Let $\Psi_I$ be the gravitino in type IIB. The gravitino has chirality $\overline{\Gamma}\Psi_I = +\Psi_I$ since we have the equation $[Q, e_M^I] \sim \Gamma^I \Psi_M$ where $e_M^I$ is the gravity field, and hence the chirality of the supercharge $Q$ and the gravitino $\Psi_I$ are opposite. The field strength $F_5 = dC_4$ appears in the transformation of $\Psi_I$ as [27][7]

$$\{\Psi_I, Q^T \mathsf{C}\} \sim (\Gamma^{I_1 \cdots I_5} F_{I_1 \cdots I_5})\Gamma_I. \tag{2.22}$$

From this transformation law and $\overline{\Gamma}\Psi_I = +\Psi_I$, we get

$$\overline{\Gamma}\Gamma^{I_1 \cdots I_5} F_{I_1 \cdots I_5} = +\Gamma^{I_1 \cdots I_5} F_{I_1 \cdots I_5}. \tag{2.23}$$

On the other hand, in the Euclidean signature,

$$\overline{\Gamma}\Gamma^{I_1 \cdots I_{p+2}} = -i(-1)^{\frac{1}{2}(p+2)(p+3)} \cdot \frac{1}{(8-p)!} \epsilon^{I_1 \cdots I_{p+2} J_1 \cdots J_{8-p}} \Gamma_{J_1 \cdots J_{8-p}}, \tag{2.24}$$

and hence the 5-form transforms as

$$\frac{1}{5!} F_{I_1 \cdots I_5} \epsilon^{I_1 \cdots I_5 J_1 \cdots J_5} = -i F^{J_1 \cdots J_5}. \tag{2.25}$$

In the notation of differential forms, it is simply written as

$$\star F_5 = -i F_5. \tag{2.26}$$

The imaginary unit $i$ appears because we are using the Euclidean signature.

---

[7] See also page 393 [28] for helpful summary.

**T-duality:** Type II string theories have two Majorana-Weyl supercharges which we denote by $Q$ and $\tilde{Q}$. We always take $Q$ to have chirality $\overline{\Gamma}Q = -Q$. Then $\tilde{Q}$ has chirality $\overline{\Gamma}\tilde{Q} = -\tilde{Q}$ in type IIB and $\overline{\Gamma}\tilde{Q} = +\tilde{Q}$ in type IIA.

Under the T-dual in the direction $I$, the supercharge $Q$ is invariant, but the supercharge $\tilde{Q}$ before and after the T-dual are related as

$$\tilde{Q}^{\text{before}} = \Gamma^I \tilde{Q}^{\text{after}}. \tag{2.27}$$

The sign is just a convention and it could as well be $\tilde{Q}^{\text{before}} = -\Gamma^I \tilde{Q}^{\text{after}}$, but we use the above convention for definiteness. In fact, there is a symmetry in type II string theories, usually denoted by $(-1)^{F_L}$, which acts on the supercharges as

$$(-1)^{F_L}(Q) = Q, \qquad (-1)^{F_L}(\tilde{Q}) = -\tilde{Q}. \tag{2.28}$$

Then, the sign (2.27) in the T-dual can be changed by an additional operation of $(-1)^{F_L}$.

We remark that T-dual actions in two orthogonal directions $I$ and $J$ ($I \neq J$) do not commute. There is an extra sign factor $(-1)^{F_L}$ in their commutation relation.

Let us start from type IIB and type I theories and take T-duals. We use the convention of the relative sign of the two supercharges in type IIB such that the supercharge preserved in type I string theory is the linear combination

$$Q + \tilde{Q}. \tag{2.29}$$

This means the following. In type IIB, there is a $\mathbb{Z}_2$ symmetry $\Omega$, satisfying $\Omega^2 = 1$, which is the orientation reversal in the string world sheet. In type I construction, we divide the theory by this symmetry. If this $\Omega$ acts on supercharges as

$$\Omega(Q) = \tilde{Q}, \qquad \Omega(\tilde{Q}) = Q, \tag{2.30}$$

then $Q + \tilde{Q}$ is the supercharge of type I. More discussions on the symmetries $(-1)^{F_L}$ and $\Omega$ are given in Appendix A.

Consider a flat spacetime in which the directions $p+1, \cdots, 9$ are compactified on a torus $T^{9-p}$. By taking T-duals in the directions $9, 8, \cdots, p+1$ in this order, the above supercharge in type I becomes

$$Q^{(\mathrm{D}p)} := Q + \Gamma^9 \Gamma^8 \cdots \Gamma^{p+1} \tilde{Q}. \tag{2.31}$$

We use the convention that after the above T-duals, the D9-branes and O9-planes become D$p$-branes and O$p$-planes with the orientation $dx^0 \wedge \cdots \wedge dx^p$, rather than anti-D$p$-branes and anti-O$p$-planes. Then the above supercharge (2.31) is the linear combination which is preserved by the D$p$-branes and O$p$-branes with the above orientation.

We also define the RR fields such that the coupling to the D$p$-brane is given by

$$-S \supset 2\pi i \int C_{p+1}, \tag{2.32}$$

rather than $-2\pi i \int C_{p+1}$. This fixes the sign of the RR-fields.

In the above definitions of D-branes and RR-fields, the T-dual in the direction $J$ acts on RR fields as

$$(F_{k+1})_{I_1 \cdots I_k J} = (F_k)_{I_1 \cdots I_k}. \tag{2.33}$$

From this, we get

$$\star F_k = \beta F_{10-k} \implies \star F_{k+1} = \beta(-1)^{k+1} F_{9-k}. \tag{2.34}$$

Combining this with (2.26), we get

$$\star F_k = +(-1)^{\frac{1}{2}k(k+1)} i F_{10-k}. \tag{2.35}$$

This is the self-duality condition of the RR-fields.

The sign factor in (2.35) has the following explanation. For simplicity, let us consider a spacetime $X$ without orientifold. RR-fluxes are classified by K-theory elements $x \in K^i(X)$ where $i = 0$ for Type IIA and $i = 1$ for Type IIB [29]. Let $F(x)$ be RR fields corresponding to the topological class $x$. In the de-Rham cohomology, its topological class is given by $\sqrt{\hat{A}}\mathrm{ch}(x)$. In [30], a complex structure $J$ was introduced on the space of RR-fluxes given by $JF(x) = \star F(\bar{x})$, where $\bar{x}$ is the complex conjugate of $x$. The Chern character $\mathrm{ch}(x)$ has the property that its $k$-form part transforms as $\mathrm{ch}(\bar{x})|_k = (-1)^{\frac{1}{2}k(k+1)}\mathrm{ch}(x)|_k$.[8] Therefore, the complex structure $J$ is explicitly given by $J(F_{10-k}) = (-1)^{\frac{1}{2}k(k+1)} \star F_k$. The equation (2.35) just means that $J = i$. Therefore, in Euclidean signature manifolds, the space of RR-fluxes is holomorphic. (See [29, 30] for more precise meaning of this statement.) This is the self-duality condition of RR-fluxes in Euclidean signature spaces.

**RR-flux and Dirac pairing:** The Euclidean action involving RR fields $C$ is schematically of the form

$$-S_E \supset -\frac{2\pi}{2} \int dC_{p+1} \wedge \star dC_{p+1} + 2\pi i \mathsf{q}_p \int_{Z^{(p+1)}} C_{p+1}, \tag{2.36}$$

where $Z^{(p+1)}$ is the world volume of a D-brane or O-plane, and $\mathsf{q}_p$ is its charge which is $\mathsf{q}_p = 1$ for a single D$p$-brane and $\mathsf{q}_p = +2^{p-5}$ for an O$p^+$-plane. This action is schematic since we have not taken into account the self-dual equation in the sense of (2.35). However, the equation of motion

$$2\pi(-1)^{p+1} d(\star F_{p+2}) + 2\pi i \mathsf{q}_p \delta(Z^{(p+1)}) = 0 \tag{2.37}$$

is valid, where $\delta(Z^{(p+1)})$ is the delta function such that

$$\int_{Z^{(p+1)}} C_{p+1} = \int C_{p+1} \wedge \delta(Z^{(p+1)}). \tag{2.38}$$

This $\delta(Z^{(p+1)})$ is the Poincaré dual of $Z^{(p+1)}$. By using (2.35), we get

$$dF_{8-p} = +(-1)^{\frac{1}{2}(p+1)(p+2)} \mathsf{q}_p \delta(Z^{(p+1)}). \tag{2.39}$$

Notice that the imaginary unit $i$ has disappeared.

Now, let $X^{(8-p)}$ be a closed submanifold with linking number $+1$ with $Z^{(p+1)}$. Namely, let $W^{(9-p)}$ be a manifold with $\partial W^{(9-p)} = X^{(8-p)}$, and that $\int_{W^{(9-p)}} \delta(Z^{(p+1)}) = +1$. From the above, we get

$$\int_{X^{(8-p)}} F_{8-p} = +(-1)^{\frac{1}{2}(p+1)(p+2)} \mathsf{q}_p. \tag{2.40}$$

The phase ambiguity of a single D$q$-brane with $q = 6-p$ wrapping a codimension-one submanifold $Y^{(q+1)}$ of $X^{(q+2)}$ (where $q + 2 = 8 - p$) is given as

$$\exp\left(2\pi i \int_{X^{(8-p)}} F_{8-p}\right) = \exp\left(+(-1)^{\frac{1}{2}(p+1)(p+2)} 2\pi i \mathsf{q}_p\right). \tag{2.41}$$

---

[8] For $x \in K^1(X) = K^{-1}(X)$, the Chern character is defined as follows. By definition, $K^{-1}(X)$ is given by elements of $K(X \times S^1)$ whose restriction to $X \times \{p\}$ is trivial. Then we get $\mathrm{ch}(x)$ as an element of $H^{\mathrm{even}}(X \times S^1)$ which is trivial when restricted to $X \times \{p\}$. We can push forward it to $X$ by integrating over $S^1$. By this process, we get odd degree forms $\mathrm{ch}(x) \in H^{\mathrm{odd}}(X)$. In particular, the $(2n-1)$-form $\mathrm{ch}(x)|_{2n-1}$ transforms in the same way as $2n$-form $\mathrm{ch}(x)|_{2n}$ under complex conjugation of $x$.

## 2.3 O-planes and fermions

Let us consider the construction of the O$p$-plane which preserves the supercharge (2.31). The local geometry near the O$p$-plane is

$$\mathbb{R}^{p+1} \times (\mathbb{R}^{9-p}/\mathbb{Z}_2). \tag{2.42}$$

In the construction, we divide the spacetime $\mathbb{R}^{p+1} \times \mathbb{R}^{9-p}$ by the operation $\vec{z} \to -\vec{z}$, where $\vec{z} = (x^{p+1}, \cdots, x^9)$ are the coordinates orthogonal to the O$p$-plane. We also need to combine it with the orientation reversal $\Omega$ of the string world sheet. More precisely, we need an uplift to the group Spin in type IIB or Pin$^+$ in type IIA. We denote this operation by $\mathsf{R}^{(9-p)}\Omega$. In type IIB, each of $\mathsf{R}^{(9-p)}$ and $\Omega$ is well-defined independently, while in type IIA only the combination $\mathsf{R}^{(9-p)}\Omega$ is well-defined for even $p$.

Explicitly, we take it to be

$$\mathsf{R}^{(9-p)}\Omega(Q) = \Gamma^9\Gamma^8\cdots\Gamma^{p+1}\tilde{Q}, \qquad \mathsf{R}^{(9-p)}\Omega(\tilde{Q}) = \Gamma^9\Gamma^8\cdots\Gamma^{p+1}Q. \tag{2.43}$$

Applying this to (2.31) gives

$$\mathsf{R}^{(9-p)}\Omega(Q^{(\mathrm{D}p)}) = (-1)^{\frac{1}{2}p(p-1)}Q + \Gamma^9\Gamma^8\cdots\Gamma^{p+1}\tilde{Q}, \tag{2.44}$$

where we have used $(\Gamma^9\Gamma^8\cdots\Gamma^{p+1})^2 = (-1)^{\frac{1}{2}p(p-1)}$. The D$p$-brane and the O$p$-plane must preserve the same supercharge, so we have to combine $\mathsf{R}^{(9-p)}\Omega$ with $(-1)^{F_L}$ in the construction of the O$p$-plane;

$$\mathsf{R}^{(9-p)}\Omega(-1)^{\frac{1}{2}p(p-1)F_L}(Q^{(\mathrm{D}p)}) = Q^{(\mathrm{D}p)}. \tag{2.45}$$

Therefore, the $\mathbb{Z}_2$ action in the construction of the O$p$-plane is generated by

$$\mathsf{R}^{(\mathrm{O}p)} := \mathsf{R}^{(9-p)}\Omega(-1)^{\frac{1}{2}p(p-1)F_L}. \tag{2.46}$$

To compute the anomaly of the gaugino of the D$q$-brane with $q = 6 - p$, it is important to know the action of the above symmetry operation on the gaugino on the D$q$-brane. Let us place the D$q$-brane at

$$\{x^0 = \cdots = x^{p+2} = 0\}. \tag{2.47}$$

We take the orientation of this D$q$-brane as

$$dx^{p+3} \wedge dx^{p+4} \wedge \cdots \wedge dx^9. \tag{2.48}$$

By the above convention of D-branes discussed around (2.31), we see that the supercharge preserved by this D$q$-brane is given as

$$Q^{(\mathrm{D}q)} := Q + (-1)^{p+1}\Gamma^{p+2}\cdots\Gamma^1\Gamma^0\tilde{Q}. \tag{2.49}$$

Here the sign factor $(-1)^{p+1}$ appeared because

$$\begin{aligned}(dx^0 \wedge \cdots \wedge dx^{p+2}) \wedge (dx^{p+3} \wedge \cdots \wedge dx^9) = \\ (-1)^{p+1}(dx^{p+3} \wedge \cdots \wedge dx^9) \wedge (dx^0 \wedge \cdots \wedge dx^{p+2}).\end{aligned} \tag{2.50}$$

To make the notation simpler, we define

$$\Gamma_{\mathrm{R}1} = \Gamma^9\Gamma^8\cdots\Gamma^{p+1}, \qquad \Gamma_{\mathrm{R}2} = (-1)^{p+1}\Gamma^{p+2}\cdots\Gamma^1\Gamma^0, \tag{2.51}$$

which satisfy

$$(\Gamma_{R1})^2 = (-1)^{\frac{1}{2}p(p-1)}, \quad (\Gamma_{R2})^2 = (-1)^{p+1}(-1)^{\frac{1}{2}p(p-1)}, \quad \Gamma_{R1}\Gamma_{R2} = (-1)^{p+1}\Gamma_{R2}\Gamma_{R1}. \quad (2.52)$$

By using them, we have $Q^{(Dq)} = Q + \Gamma_{R2}\tilde{Q}$ and

$$\begin{aligned}
R^{(Op)}(Q^{(Dq)}) &= R^{(9-p)}\Omega\left(Q + (-1)^{\frac{1}{2}p(p-1)}\Gamma_{R2}\tilde{Q}\right) \\
&= \Gamma_{R1}\tilde{Q} + (-1)^{\frac{1}{2}p(p-1)}\Gamma_{R2}\Gamma_{R1}Q \\
&= (-1)^{\frac{1}{2}p(p-1)}\Gamma_{R2}\Gamma_{R1}(Q + \Gamma_{R2}\tilde{Q}),
\end{aligned} \quad (2.53)$$

so we get

$$R^{(Op)}(Q^{(Dq)}) = (-1)^{\frac{1}{2}p(p-1)}\Gamma_{R2}\Gamma_{R1}Q^{(Dq)}. \quad (2.54)$$

By using this result, we want to see how the gaugino $\lambda$ on the D$q$-brane transforms under the transformation $R^{(Op)} = R^{(9-p)}\Omega(-1)^{\frac{1}{2}p(p-1)F_L}$. Let us consider the supersymmetry transformation

$$[Q^{(Dq)}, \phi^I] \sim \Gamma^I \lambda, \quad (2.55)$$

where $\phi^I$ are world volume scalar fields of the D$q$-brane and the index $I$ runs over the normal directions to the world volume. The supercharge $Q^{(Dq)}$ transforms as in (2.54). On the other hand, the scalars $\phi^I$ are proportional to the positions of the D$q$-brane, and hence they transform as

$$R^{(Op)}(\phi_I) = \begin{cases} -\phi^I & I = (p+1), (p+2), \\ \phi^I & I = 0, 1, \cdots, p. \end{cases} \quad (2.56)$$

Therefore, the gaugino on the D$q$-brane transforms as

$$\begin{aligned}
R^{(Op)}(\lambda) &= -(-1)^{\frac{1}{2}p(p-1)}\Gamma_{R2}\Gamma_{R1}\lambda \\
&= -(-1)^{\frac{1}{2}p(p-1)}(\Gamma^0 \cdots \Gamma^p)(\Gamma^{p+1}\Gamma^{p+2})^2(\Gamma^{p+3} \cdots \Gamma^9)\lambda \\
&= -(-1)^{\frac{1}{2}p(p-1)}i\Gamma^{p+1}\Gamma^{p+2}\overline{\Gamma}\lambda,
\end{aligned} \quad (2.57)$$

where $\overline{\Gamma} = i^{-5}\Gamma^0 \cdots \Gamma^9$. This is the transformation of the gaugino in $q + 1$ dimensions placed at (2.47).

We are actually interested in the D$q$-brane which is wrapped in a codimension-1 submanifold $Y^{(q+1)}$ of the $(q + 2)$-dimensional space $X^{(q+2)}$ given by

$$X^{(q+2)} = \{(\vec{0}, \vec{z}) \in \{0\} \times \mathbb{R}^{9-p}/\mathbb{Z}_2; \; |\vec{z}| = r\}, \quad (2.58)$$

where $\vec{z} = (x^{p+1}, \cdots, x^9)$ and $r$ is an arbitrary positive constant. Before the division by $\mathbb{Z}_2$, the gaugino is a section of the spin bundle of $\mathbb{R}^{10}$ which contains the spin bundles of both the tangent and normal bundles of the worldvolume. We trivialize this total spin bundle by using the flat metric of $\mathbb{R}^{10}$, no matter how the worldvolume is curved. Then, the $\mathbb{Z}_2$ action relates the gauginos at the positions $\vec{z}$ and $-\vec{z}$ by

$$R^{(Op)}(\lambda(\vec{z})) = -(-1)^{\frac{1}{2}p(p-1)}i\Gamma^x\Gamma^y\overline{\Gamma}\lambda(-\vec{z}), \quad (2.59)$$

where $\Gamma^x$ is the gamma matrix in the radial direction $\vec{e}_x = \vec{z}/|\vec{z}|$, $\Gamma^y$ is the gamma matrix in the direction $\vec{e}_y$ which is normal to $Y^{(q+1)}$ in $X^{(q+2)}$. This equation follows from (2.57) by

slightly moving the D$q$-brane from $(x^{p+1}, x^{p+2}) = (0,0)$ to $(x^{p+1}, x^{p+2}) = (r, 0)$ and considering the point $\vec{z} = (r, 0, \cdots, 0)$. In this case we have $\Gamma^x = \Gamma^{p+1}$ and $\Gamma^y = \Gamma^{p+2}$.[9] At the point $\vec{z} = (-r, 0, \cdots, 0)$ we have $\Gamma^x = -\Gamma^{p+1}$ and $\Gamma^y = -\Gamma^{p+2}$, but the product is the same; $(-\Gamma^{p+1})(-\Gamma^{p+2}) = \Gamma^{p+1}\Gamma^{p+2}$. In the orientifold, we impose the identification under $R^{(Op)}$ as

$$\lambda(\vec{z}) \sim -(-1)^{\frac{1}{2}p(p-1)} i \Gamma^x \Gamma^y \overline{\Gamma} \lambda(-\vec{z}). \tag{2.60}$$

**Trivialization of the normal bundle:**    In the above computation, we must notice the following point. The gaugino is a section of the total spin bundle $S(TY^{(q+1)} \oplus NY^{(q+1)})$ where $TY^{(q+1)}$ is the tangent bundle and $NY^{(q+1)}$ is the normal bundle. The normal bundle is the bundle of the R-symmetry from the point of view of the worldvolume theory. The sum $TY^{(q+1)} \oplus NY^{(q+1)}$ is just the tangent bundle of the ten dimensional bulk restricted to the worldvolume $Y^{(q+1)}$. The result (2.60) is valid in the trivialization of the bundle which follows naturally from the flat space structure of the bulk spacetime $\mathbb{R}^{10}$. However, to compare with the computation of the $\eta$-invariant discussed below, we want to trivialize the normal bundle $NY^{(q+1)}$ rather than the sum $TY^{(q+1)} \oplus NY^{(q+1)}$ so that the R-symmetry plays no role in the computation of the $\eta$-invariant.

The D$q$-brane is moved within the plane $x^0 = \cdots = x^p = 0$, so these directions are always trivial and we neglect them. Then the normal bundle $NY^{(q+1)}$ is effectively spanned by the $x$-direction $\vec{e}_x = \vec{z}/|\vec{z}|$ and the $y$-direction $\vec{e}_y$ introduced above. We take the orthonormal frame $(\vec{e}_x, \vec{e}_y)$ to trivialize $NY^{(q+1)}$.

For illustration, let us see the effect of trivialization of the normal bundle when the D$q$-brane is at $(x^{p+1}, x^{p+2}) = (r\cos\theta, r\sin\theta)$ and extending to other directions $(x^{p+3}, \cdots, x^9)$. On the worldvolume, consider a point $(x^{p+3}, \cdots, x^9) = 0$. At that point, the orthonormal vectors in the $x$ and the $y$ directions are given by

$$\vec{e}_x = (\cos\theta, \sin\theta, 0\cdots, 0), \qquad \vec{e}_y = (-\sin\theta, \cos\theta, 0\cdots, 0). \tag{2.61}$$

Then the gaugino $\lambda'$ after trivializing $NY^{(q+1)}$ by the frame $(\vec{e}_x, \vec{e}_y)$ is given by

$$\lambda' = \exp\left(\frac{\theta}{2}\Gamma^{p+1}\Gamma^{p+2}\right)\lambda. \tag{2.62}$$

---

[9] The reason that $\Gamma^y = \Gamma^{p+2}$ instead of $\Gamma^y = -\Gamma^{p+2}$ is seen as follows. We have to recall the process of how to compute the ambiguity of the coupling $2\pi i \int C_{q+1}$ with the RR-field. It is simpler to consider the covering space $\tilde{X}^{(q+2)} \cong S^{(q+2)}$ so that all (sub)manifolds are orientable. Not only orientable, we actually need to orient them to compute the coupling. Our computation of the RR-flux $\int_{X^{(q+2)}} F_{q+2}$ in Sec. 2.2 implicitly used the convention that this sphere $S^{(q+2)}$ is oriented by a $(q + 2)$-form $\omega_{(q+2)}$ such that $dx \wedge \omega_{(q+2)} = dx^{p+1} \wedge \cdots \wedge dx^9$ where $x = |\vec{z}|$. Moreover, the orientation of $Y^{(q+1)}$ must be given by a $(q + 1)$-form $\omega_{(q+1)}$ such that $dy \wedge \omega_{(q+1)} = \omega_{(q+2)}$. This orientation is required by the Stokes theorem $\int_{Y^{(q+1)}} C_{q+1} = \int_{X'^{(q+2)}} F_{q+2}$, where $\partial X'^{(q+2)} = Y^{(q+1)}$, and the fact that $dy$ is in the outgoing direction as taken in Sec. 2.1. We also have to recall that the D$q$-brane is oriented by (2.48). Therefore, at the point $\vec{z} = (r, 0, \cdots, 0)$, we have $\omega_{(q+1)} = dx^{p+3} \wedge dx^{p+4} \wedge \cdots \wedge dx^9$. Combining this with $dx = dx^{p+1}$ which gives $\omega_{(q+2)} = dx^{p+2} \wedge dx^{p+3} \wedge \cdots \wedge dx^9$ at $\vec{z} = (r, 0, \cdots, 0)$, we finally get $dy = dx^{p+2}$. This is the reason that $\Gamma^y = \Gamma^{p+2}$.

In the case of type IIA, the manifold $X^{(q+2)} = S^{(q+2)}/\mathbb{Z}_2 = \mathbb{RP}^{(q+2)}$ is not orientable. However, the RR field changes the sign under $\mathbb{Z}_2$ as $C_{q+1} \to -C_{q+1}$ in addition to the usual transformation under $\vec{z} \to -\vec{z}$. Therefore, the integrations of $C_{q+1}$ and $F_{q+2}$ are well-defined without orientation. Their values are half of the integration of them in the oriented double cover such as $S^{(q+2)}$.

The process of going from $(q+1)$-dimensional $Y^{(q+1)}$ to $(q+2)$-dimensional $X^{(q+2)}$ must be done in the same way in the computation of both the RR-coupling and the $\eta$ invariant. Therefore, the directions $x$ and $y$ determined above from the consideration of the RR-coupling are exactly what should be used in the later computation of the $\eta$ invariant.

The points $\theta = 0$ and $\theta = \pi$ are identified as

$$\lambda'(\theta = \pi) = \exp\left(\frac{\pi}{2}\Gamma^{p+1}\Gamma^{p+2}\right)\lambda(\theta = \pi) \tag{2.63}$$

$$\sim \Gamma^{p+1}\Gamma^{p+2}\left(-(-1)^{\frac{1}{2}p(p-1)}i\Gamma^{p+1}\Gamma^{p+2}\overline{\Gamma}\lambda(\theta = 0)\right) \tag{2.64}$$

$$= (-1)^{\frac{1}{2}p(p-1)}i\overline{\Gamma}\lambda'(\theta = 0), \tag{2.65}$$

where we have used (2.60).

The $\overline{\Gamma}$ is the chirality operator of the gaugino. We have the chirality condition $\overline{\Gamma}\lambda = +\lambda$ as discussed around (2.21).

**Descent structure of fermions:** To compute the anomaly by using the process described in Sec. 2.1, we want to uplift the situation from $Y^{(q+1)}$ to

$$W^{(q+3)} = \{(\vec{0}, \vec{z}) \in \{0\} \times \mathbb{R}^{9-p}/\mathbb{Z}_2; \; |\vec{z}| \leq r\}. \tag{2.66}$$

We remark that the "fermions" which appear in the following discussions are just mathematically introduced for the purpose of computation of the $\eta$-invariant, and not physical fields in the string theory.

Let us consider a fermion on $W^{(q+3)}$ with gamma matrices $\Gamma_W^I$ ($I = p+1, \cdots, 9$) and the chirality operator $\overline{\Gamma}_W$ which transforms as

$$R^{(Op)}(\lambda_{(q+3)}(\vec{z})) = +(-1)^{\frac{1}{2}p(p-1)}\overline{\Gamma}_W \lambda_{(q+3)}(-\vec{z}). \tag{2.67}$$

The insertion of $\overline{\Gamma}_W$ in this equation needs explanation. On $W^{(q+3)}$ we use the Dirac operator in flat space[10]

$$\mathcal{D}_W = i \sum_{I=p+1}^{9} \Gamma_W^I \partial_I. \tag{2.68}$$

This Dirac operator must transform covariantly under $R^{(Op)}$. The spacetime coordinates transform as $x^I \to -x^I$ for $I = p+1, \cdots, 9$, and hence the derivatives with respect to them transform as $\partial_I \to -\partial_I$. The $\overline{\Gamma}_W$, which anti-commutes with the gamma matrices $\Gamma_W^I$, was inserted so that

$$\mathcal{D}_W\left[R^{(Op)}(\lambda_{(q+3)})\right] = R^{(Op)}(\mathcal{D}_W \lambda_{(q+3)}). \tag{2.69}$$

Thus the Dirac operator $\mathcal{D}_W$ is equivariant under the $\mathbb{Z}_2$ operation $R^{(Op)}$.

Following the process in Sec. 2.1, we first reduce it to

$$X^{(q+2)} = \{(\vec{0}, \vec{z}) \in \{0\} \times \mathbb{R}^{9-p}/\mathbb{Z}_2; \; |\vec{z}| = r\}. \tag{2.70}$$

This is done by the projection

$$\lambda_{(q+2)} = \left(\frac{1+\overline{\Gamma}_W}{2}\right)\lambda_{(q+3)}. \tag{2.71}$$

It transforms as

$$R^{(Op)}(\lambda_{(q+2)}(\vec{z})) = +(-1)^{\frac{1}{2}p(p-1)}\lambda_{(q+2)}(-\vec{z}), \tag{2.72}$$

---

[10] More precisely, we use some Weyl rescaling of it near the boundary $X^{(q+2)}$ so that the metric on $W^{(q+3)}$ near $X^{(q+2)}$ is of the product form $(-\epsilon, 0] \times X^{(q+2)}$.

because we have $\overline{\Gamma}_W = +1$ after the projection.

Next, we reduce the fermion $\lambda_{(q+2)}$ on $X^{(q+2)}$ to the chiral fermion $\lambda_{(q+1)}$ on $Y^{(q+1)}$. The chirality operator is $\overline{\Gamma}_Y = \Gamma_X^y = -i\Gamma_W^x \Gamma_W^y$ as discussed in Sec. 2.1, and we identify it with the chirality operator $\overline{\Gamma}$ which is relevant for the worldvolume gaugino. The chiral fermion satisfies $\overline{\Gamma}_Y \lambda_{(q+1)} = +\lambda_{(q+1)}$. The transformation is again

$$R^{(Op)}(\lambda_{(q+1)}(\vec{z})) = +(-1)^{\frac{1}{2}p(p-1)}\lambda_{(q+1)}(-\vec{z}), \tag{2.73}$$

and we impose the identification $R^{(Op)}(\lambda_{(q+1)}(\vec{z})) \sim \lambda_{(q+1)}(\vec{z})$. However, we have to notice that the above transformation is valid in the trivialization of the spin bundle which naturally follows from the flat space structure of $W^{(q+3)}$. For the process of Sec. 2.1 to be valid, we have to take the frame associated to $(\vec{e}_x, \vec{e}_y)$. For example, by restricting attention to $(x^{p+1}, x^{p+2}) = (\cos\theta, \sin\theta)$, we change the frame as

$$\begin{aligned}
\lambda'_{(q+1)} &= \exp\left(\frac{\theta}{2}\Gamma_W^{p+1}\Gamma_W^{p+2}\right)\lambda_{(q+1)} = \exp\left(\frac{\theta}{2}i\overline{\Gamma}_Y\right)\lambda_{(q+1)} \\
&= \exp\left(\frac{\theta}{2}i\right)\lambda_{(q+1)},
\end{aligned} \tag{2.74}$$

where we have used the chirality condition $\overline{\Gamma}_Y\lambda = +\lambda$. Then the identification is

$$\lambda'_{(q+1)}(\theta = \pi) \sim +i(-1)^{\frac{1}{2}p(p-1)}\lambda'_{(q+1)}(\theta = 0). \tag{2.75}$$

This reproduces (2.65) after using $\overline{\Gamma}\lambda = +\lambda$. Therefore, the fermion $\lambda'_{(q+1)}$ transforms in the same way as the gaugino $\lambda'$. We conclude that (2.67) is the correct transformation for the index computation.

## 2.4 The $\eta$ invariant and the anomaly cancellation

**Setup:** Recall that we are interested in the following situation. The 10d manifold is

$$M^{(10)} = \mathbb{R}^{p+1} \times (\mathbb{R}^{9-p}/\mathbb{Z}_2). \tag{2.76}$$

The O$p$-plane is located at

$$Z^{(p+1)} = \{x^{p+1} = \cdots = x^9 = 0\} = \mathbb{R}^{p+1} \times \{0\}. \tag{2.77}$$

The flux of $F_{8-p}$ through

$$X^{(8-p)} = \mathbb{RP}^{8-p} = \{(\vec{0}, \vec{z}) \in \{0\} \times \mathbb{R}^{9-p}/\mathbb{Z}_2; \, |\vec{z}| = r\} \tag{2.78}$$

is given by $q_p = +2^{p-5}$ for O$p^+$-plane. We also often use $q = 6-p$. The $\eta$ invariant on $X^{(8-p)}$ is relevant for the D$q$-brane wrapped on a codimension-1 submanifold $Y^{(q+1)}$ of $X^{(q+2)}$.

**Computation of the $\eta$ invariant:** Let us compute the $\eta$ invariant following the strategy discussed in the introduction. The part $\mathbb{R}^{p+1}$ in $M^{(10)}$ plays no role and hence we neglect it. To compute the $\eta$ invariant, we consider the manifold

$$T^{q+3}/\mathbb{Z}_2, \tag{2.79}$$

where $T^{q+3}$ is the $(q+3)$-dimensional torus. We denote the fiber of the (s)pin bundle on $T^{q+3}$ as $S$, and a (trivial) bundle on $T^{q+3}$ as $E$. On the bundle whose total space is $T^{q+3} \times S \otimes E$, we act $\mathbb{Z}_2$ by

$$R^{(Op)} : (\vec{z}, s) \mapsto (-\vec{z}, +(-1)^{\frac{1}{2}p(p-1)}\overline{\Gamma}_W s), \tag{2.80}$$

where $\vec{z} = (x^{p+1}, x^{p+2}, \cdots, x^9)$ are the coordinates of $T^{q+3}$ and $s$ is the coordinate of the fiber $S \otimes E$. The reason that we have introduced the bundle $E$ is that the gauginos take values in the bundle $S \otimes E$ for some $E$ because of the symmetries $\Omega$ and $(-1)^{F_L}$ as well as the normal bundle. The factor $+(-1)^{\frac{1}{2}p(p-1)} \overline{\Gamma}_W$ acting on $s$ is the one which appears in (2.67).

We consider the Dirac operator $\mathcal{D}_W = i \Gamma^I \partial_I$ which acts on sections of the bundle

$$(T^{q+3} \times S \otimes E)/\mathbb{Z}_2, \tag{2.81}$$

or in other words, $\mathcal{D}_W$ is equivariant under the $\mathbb{Z}_2$ transformation. Near each fixed point of $T^{q+3}/\mathbb{Z}_2$, the local geometry is the same as $\mathbb{R}^{q+3}/\mathbb{Z}_2$ which appears in $M^{(10)}$. Let $B_{\text{total}} = B_1 \sqcup B_2 \sqcup \cdots \sqcup B_{2^{q+3}}$ be the disjoint union of small balls around each of the fixed points of $T^{q+3}/\mathbb{Z}_2$. Applying the APS index theorem to

$$W' = (T^{q+3}/\mathbb{Z}_2) \setminus B_{\text{total}}, \tag{2.82}$$

we have

$$\text{index}\,\mathcal{D}_{W'} = \eta(\mathcal{D}_{\partial W'}) = \eta(\mathcal{D}_{\overline{\partial B_{\text{total}}}}) = -\eta(\mathcal{D}_{\partial B_{\text{total}}}) = -2^{q+3}\eta(\mathcal{D}_{X^{(q+2)}}). \tag{2.83}$$

Here the overline on $\overline{\partial B_{\text{total}}}$ means the opposite spin/pin structure from that on $\partial B_{\text{total}}$ whose details are described in [31,32]. It is a generalization of the orientation flip, but this operation is nontrivial even on non-orientable manifolds when there are pin structures. The equality $\eta(\mathcal{D}_{\overline{\partial B_{\text{total}}}}) = -\eta(\mathcal{D}_{\partial B_{\text{total}}})$ is a simple consequence of the fact that we represent the Dirac operator $\mathcal{D}_W$ near the boundary as $\Gamma_W^x(\partial_x + \mathcal{D}_X)$ as discussed in Sec. 2.1, and the operator $\mathcal{D}_X$ changes the sign when we change the normal direction as $x \to x' = -x$. In the last step, we have also used the fact that each $\partial B_k$ ($k = 1, \cdots, 2^{q+3}$) is a copy of $X^{(q+2)}$.

On the other hand, the index $\text{index}\,\mathcal{D}_{W'}$ is simply computed from the number of solutions $\mathcal{D}_{T^{q+3}}\lambda_{(q+3)} = 0$ which satisfy $\lambda_{(q+3)}(\vec{z}) = +(-1)^{\frac{1}{2}p(p-1)}\overline{\Gamma}_W\lambda_{(q+3)}(-\vec{z})$. In fact, the APS boundary condition is such that the solutions on $W = (T^{q+3}/\mathbb{Z}_2)\setminus B$ are extended to $T^{q+3}/\mathbb{Z}_2$ without any divergence at the fixed points.[11] In other words, the index on $W = (T^{q+3}/\mathbb{Z}_2) \setminus B$ is the same as the index on $T^{q+3}$ of the modes which are invariant under $\mathbb{Z}_2$.

The Dirac equation on $T^{q+3}$ is satisfied if and only if $\lambda_{(q+3)}$ is constant. Therefore, we simply need to count the number of components satisfying

$$\lambda_{(q+3)} = +(-1)^{\frac{1}{2}p(p-1)}\overline{\Gamma}_W\lambda_{(q+3)}. \tag{2.84}$$

Then the index $\text{index}\,\mathcal{D}_W$ is given by

$$\text{index}\,\mathcal{D}_W = +(-1)^{\frac{1}{2}p(p-1)} \times \frac{1}{2} \times 4 \times 8 = +(-1)^{\frac{1}{2}p(p-1)} \cdot 2^4. \tag{2.85}$$

This formula is understood as follows. First, on the D$q$-brane, the total number of complex components of the gaugino is 8, since the 10d Majorana-Weyl fermion has 16 real or 8 complex components. Next, in the process of going from dimension $d = q + 1$ for $\lambda_{(q+1)}$ to dimension $d + 2 = q + 3$ for $\lambda_{(q+3)}$, the number of components is increased by a factor of $4 = 2 \times 2$ because $\lambda_{(q+1)}$ is obtained from $\lambda_{(q+3)}$ by two projections $\frac{1}{2}(1 + \overline{\Gamma}_W)$ and $\frac{1}{2}(1 + \overline{\Gamma}_Y)$. Finally, the number of fermions satisfying (2.84) is half of these components in $(q + 3)$ dimensions, and all of them have chirality $\overline{\Gamma}_W = +(-1)^{\frac{1}{2}p(p-1)}$. Thus we obtain (2.85).

Therefore, from (2.83) we get

$$\eta(\mathcal{D}_{X^{(q+2)}}) = -(-1)^{\frac{1}{2}p(p-1)} \cdot 2^{p-5}. \tag{2.86}$$

---

[11] In more detail, this argument requires some Weyl rescaling near the boundary. For the application of APS index theorem, we take the metric near the boundary $\partial W$ as the product form $(-\epsilon, 0] \times \partial W$. On the other hand, in the extension of the solution from $(T^{q+3}/\mathbb{Z}_2) \setminus B$ to $T^{q+3}/\mathbb{Z}_2$, the metric which follows from the flat metric on $T^{q+3}$ is more natural. These two metrics are related by Weyl transformation.

**Anomaly cancellation:**    By the above computation, we obtain the anomaly from the gaugino as

$$\exp(-2\pi i \eta(\mathcal{D}_{X^{(q+2)}})) = \exp\left(+(-1)^{\frac{1}{2}p(p-1)}2\pi i \cdot 2^{p-5}\right). \tag{2.87}$$

On the other hand, the RR coupling (2.41) gives

$$\exp\left(2\pi i \int_{X^{(q+2)}} F_{q+2}\right) = \exp\left(+(-1)^{\frac{1}{2}(p+1)(p+2)}2\pi i \cdot 2^{p-5}\right), \tag{2.88}$$

where we have used $\mathsf{q}_p = +2^{p-5}$ for O$p^+$.    By a simple computation we get $(-1)^{\frac{1}{2}(p+1)(p+2)} = -(-1)^{\frac{1}{2}p(p-1)}$. Therefore, the above two factors cancel each other to make the partition function of the D$q$-brane well-defined. That was what we wanted to show.

# 3    Relations to symmetry-protected topological phases

We used the $\eta$ invariants in $(q+2)$ dimensions to capture global anomalies of fermions on the worldvolume of a D$q$-brane. According to [15, 31, 32], such anomalies are controlled by cobordism invariants in $(q+2)$ dimensions, which also classify the interacting fermionic symmetry protected phases in the said dimensions. We will study these cobordism invariants. In this section we do not carefully keep the precise sign factors, since they have already been treated in detail in the previous section.

For simplicity, we only consider spacetime manifolds of the form $\mathbb{R}^{8-q} \times X^{q+2}$, and discuss the $(q+2)$-dimensional cobordism invariants. This is not quite general in the context of string theory, where we can also consider nontrivial topologies of the normal directions to $X^{q+2}$. In fact, nontrivial normal bundles can produce anomalies of the worldvolume fermions which are cancelled by a shift of the quantization condition of the RR fluxes [29].[12]

We include our discussions on the restricted set of worldvolumes with trivial normal bundles below, since it illustrates various variants of (s)pin bordism groups in various dimensions which were discussed in Sec. 8 of [15] and will be reviewed in the next subsection. Namely, we will see that we are considering a fermionic system in which the Lorentz group is uplifted to the symmetry group which is an extension of the spin group as follows:

| string theory | IIA | IIB | IIA | IIB |
|---|---|---|---|---|
| projective space | $\mathbb{RP}^{4m}$ | $\mathbb{RP}^{4m+1}$ | $\mathbb{RP}^{4m+2}$ | $\mathbb{RP}^{4m+3}$ |
| structure group | pin$^+$ | spin$^{\mathbb{Z}_4}$ | pin$^-$ | spin$\times\mathbb{Z}_2$ |
| uniform notation in Sec. 3.1 | spin[3] | spin[2] | spin[1] | spin[4] |

$$\tag{3.1}$$

Here $\mathrm{Spin}^{\mathbb{Z}_4}(d) = (\mathrm{Spin}(d)\times\mathbb{Z}_4)/\mathbb{Z}_2$, where the generator of the $\mathbb{Z}_2$ is the diagonal combination of the $(-1)^F \in \mathrm{Spin}(d)$ and $\exp(\pi i) \in \mathbb{Z}_4 \subset \mathrm{U}(1)$.

Before proceeding, we note that the relation between the fermionic symmetry protected topological phases and the D-brane/O-plane systems was discussed from a different perspective by Ryu and Takayanagi [33, 34]. Their aim was more about reproducing the periodic table of free fermionic classifications.

---

[12] Using the $\eta$ invariant, the anomalies described in Sec. 4 of [29] can be seen as follows. Let us consider type IIA string theory with a D$q$-brane whose worldvolume is $Y^{(q+1)}$. To consider the anomaly, we uplift the fermions to $(q+2)$-dimensional space $X^{(q+2)}$. On this manifold, there is also the normal bundle whose fiber is $\mathbb{R}^{(8-q)}$. When the manifold $X^{(q+2)}$ is orientable and even-dimensional as in the case considered in [29], the $\eta$ invariant of the Dirac operator $\mathcal{D}_X$ is given by $\exp(-2\pi i \eta(\mathcal{D}_X)) = (-1)^{\mathrm{index}\,\mathcal{D}_X}$, where index $\mathcal{D}_X$ here is the mod 2 index of $\mathcal{D}_X$ defined by using the reality condition (i.e. Majorana condition) of the spinors including the normal as well as the tangent bundles. When the normal bundle is nontrivial, this mod 2 index can have nontrivial values and hence we get a nontrivial value for the anomaly $(-1)^{\mathrm{index}\,\mathcal{D}_X}$. In this particular case, the global anomalies are just sign factors.

## 3.1 Extensions of the spin group

We can construct a class of extensions of the $d$-dimensional spin group $\mathrm{Spin}(d)$. Those extensions depend on an integer mod 4, which we denote as $k$. Let us denote that extended group as $\mathrm{Spin}(d;k)$. They are constructed as follows.

We first let $k$ be a positive integer. We then consider a subgroup $\mathrm{Spin}(d;k) \subset \mathrm{Spin}(d+k)$ such that the elements of this subgroup commute with $\mathrm{Spin}(k) \subset \mathrm{Spin}(d+k)$. In other words, if we act $\mathrm{Spin}(d;k)$ on $\mathbb{R}^{d+k} = \mathbb{R}^d \oplus \mathbb{R}^k$ as the $\mathrm{SO}(d+k)$ matrix after the reduction $\mathrm{Spin}(d+k) \to \mathrm{SO}(d+k)$, we demand that these elements should act on the part $\mathbb{R}^k$ as $\pm I_k$, where $I_k$ is the unit $k \times k$ matrix. Because $-I_k$ is allowed, $\mathrm{Spin}(d;k)$ is twice as big as $\mathrm{Spin}(d)$, roughly speaking. By considering each case $k = 1, 2, 3, 4$ explicitly, one can check that we have

$$
\mathrm{Spin}(d;1) = \mathrm{Pin}^-(d), \quad \mathrm{Spin}(d;2) = (\mathrm{Spin}(d) \times \mathbb{Z}_4)/\mathbb{Z}_2,
$$
$$
\mathrm{Spin}(d;3) = \mathrm{Pin}^+(d), \quad \mathrm{Spin}(d;4) = \mathrm{Spin}(d) \times \mathbb{Z}_2. \tag{3.2}
$$

One can also check that there are isomorphisms $\mathrm{Spin}(d;k+4) \cong \mathrm{Spin}(d;k)$, and hence $\mathrm{Spin}(d;k)$ only depends on $k$ mod 4. Using this property, we generalize our definition of $\mathrm{Spin}(d;k)$ to arbitrary integral $k$.

There are natural homomorphisms

$$
\rho_1 : \mathrm{Spin}(d;k) \to \mathrm{O}(d), \qquad \rho_2 : \mathrm{Spin}(d;k) \to \mathbb{Z}_2. \tag{3.3}
$$

These homomorphisms are defined by first projecting from $\mathrm{Spin}(d+k) \to \mathrm{SO}(d+k)$ and then considering the induced action on $\mathbb{R}^d \subset \mathbb{R}^{d+k}$ and $\mathbb{R}^k \subset \mathbb{R}^{d+k}$ respectively. These homomorphisms allow us to put $G = \mathrm{Spin}(d;k)$ into the following commutative diagram:

$$
\begin{array}{ccccccccc}
0 & \to & \mathrm{Spin}(d) & \to & G & \xrightarrow{\rho_2} & \mathbb{Z}_2 & \to & 0 \\
 & & \downarrow & & \downarrow{\scriptstyle\rho_1} & & & & \\
 & & \mathrm{SO}(d) & \to & \mathrm{O}(d), & & & &
\end{array} \tag{3.4}
$$

where the first line is an exact sequence. In fact $\mathrm{Spin}(d;k)$ for $k = 1, 2, 3, 4$ exhaust all such $G$ which fit in this diagram.

Let us call manifolds with the structure group $\mathrm{Spin}(\bullet;k)$ as spin[$k$]-manifolds.[13] In the bordism group $\Omega_d^{\mathrm{spin}[k]}$ of $d$-dimensional manifolds with spin[$k$]-structure, there is a homomorphism

$$
s : \Omega_d^{\mathrm{spin}[k]} \to \Omega_{d-1}^{\mathrm{spin}[k+1]}, \tag{3.5}
$$

called the Smith homomorphism, defined as follows.[14]

On a spin[$k$]-manifold $M_d$, we have a $\mathbb{Z}_2$-bundle which is constructed from the $\mathrm{Spin}(d;k)$ bundle and the homomorphism $\rho_2 : \mathrm{Spin}(d;k) \to \mathbb{Z}_2$. In general, a $\mathbb{Z}_2$-bundle is classified by an element $a \in H^1(M_d, \mathbb{Z}_2)$ representing the holonomy of the $\mathbb{Z}_2$ bundle. Moreover, any element $a \in H^1(M_d, \mathbb{Z}_2)$ has a Poincaré dual submanifold $N_{d-1} \subset M_d$ [35]. To construct this Poincaré dual, we consider a real line bundle $L$ on $M_d$ associated to the principal $\mathbb{Z}_2$ bundle. We take a generic section $f$ of this real line bundle $L$, and define $N_{d-1} = \{f = 0\}$ which is smooth if $f$ is generic enough. One can see that the differential $df$ gives an isomorphism

$$
L|_{N_{d-1}} \cong N(N_{d-1}), \tag{3.6}
$$

---

[13] In general, given a real bundle $\xi$ over $BG$, one can define the spin structure twisted by $\xi$ on $M$ to be a map $f : M \to BG$, i.e. a $G$-bundle, together with a spin structure of $TM \oplus f^*(\xi)$. There is a corresponding bordism group usually denoted by $\Omega_d^{\mathrm{spin}}(BG, \xi)$ and called as the twisted spin bordism groups. See Chapter 4.4.3 of [25] for details on the twisted spin bordisms. Our spin[$k$] structure corresponds to taking $G = \mathbb{Z}_2$ and choosing $\xi$ to be $L^{\oplus k}$ where $L$ is the real line bundle over $B\mathbb{Z}_2 = \mathbb{RP}^\infty$ whose $w_1$ is the generator of $H^1(\mathbb{RP}^\infty, \mathbb{Z}_2)$. In their notation $\Omega_d^{\mathrm{spin}[k]} = \Omega_d^{\mathrm{spin}}(B\mathbb{Z}_2, L^{\oplus k})$.

[14] For general discussion of Smith homomorphisms in various cobordism theories, see Chapter 3.3 of [25].

where $N(N_{d-1})$ is the normal bundle of $N_{d-1}$ in $M_d$. Then we get

$$TM_d|_{N_{d-1}} \cong TN_{d-1} \oplus L|_{N_{d-1}}. \tag{3.7}$$

From the definition of the $\mathrm{Spin}(d;k)$ groups, it follows that the $\mathrm{Spin}(d;k)$ structure on $TN_{d-1} \oplus L|_{N_{d-1}}$ is the same as the $\mathrm{Spin}(d-1;k+1)$ structure on $TN_{d-1}$. In fact, on a general manifold $M$, a $\mathsf{spin}[k]$ structure on $TM$ is a spin structure on $TM \oplus L^{\oplus k}$. Therefore, we get a map

$$s : \Omega_d^{\mathsf{spin}[k]} \ni [M_d] \mapsto [N_{d-1}] \in \Omega_{d-1}^{\mathsf{spin}[k+1]}. \tag{3.8}$$

We will use this homomorphism in later discussions.

## 3.2 Properties of (s)pinors on $\mathbb{RP}^n$

Now we consider (s)pin structures or its generalizations $\mathsf{spin}[k]$ structures on $\mathbb{RP}^n$. We will be brief; for more mathematical details, the readers should consult [36] for a standard account of pin structures in low dimensions, [37,38] for those in general dimensions, and [25] for the $\eta$ invariants of $\mathbb{RP}^n$ and other fixed-point-free quotients of spheres.

First, let us recall that the total Stiefel-Whitney class of $\mathbb{RP}^n$ is given by

$$w(\mathbb{RP}^n) = (1 + a)^{n+1}, \tag{3.9}$$

where $a$ is the generator of $H^1(\mathbb{RP}^n, \mathbb{Z}_2)$. In particular, we have

$$w_1 = (n+1)a, \qquad w_2 = \frac{(n+1)n}{2}a^2. \tag{3.10}$$

Therefore $\mathbb{RP}^{2\ell}$ is non-orientable while $\mathbb{RP}^{2\ell+1}$ is orientable.

For orientable manifolds, recall that the existence of the spin structure is equivalent to $w_2 = 0$, and the existence of the $\mathsf{spin}^c$ structure is equivalent to the condition that $w_2$ is a mod-2 reduction of an integral class. For non-orientable manifolds, recall that the existence of a $\mathsf{pin}^+$ structure or a $\mathsf{pin}^-$ structure is equivalent to $w_2 = 0$ and $w_2 + w_1^2 = 0$ respectively, and that the existence of a $\mathsf{pin}^c$ structure is equivalent to the condition that $w_2$ is a mod-2 reduction of an integral class.

We note that $a$ is the first Stiefel-Whitney class of the tautological real line bundle $L$ over $\mathbb{RP}^n$, and $a^2$ is the first Chern class of $L \otimes \mathbb{C}$. Therefore, every $\mathbb{RP}^{2\ell}$ is $\mathsf{pin}^c$ and $\mathbb{RP}^{2\ell+1}$ is $\mathsf{spin}^c$. In more detail, $\mathbb{RP}^{4m+2}$ is $\mathsf{pin}^-$, $\mathbb{RP}^{4m+3}$ is spin, $\mathbb{RP}^{4m}$ is $\mathsf{pin}^+$, and $\mathbb{RP}^{4m+1}$ is $\mathsf{spin}^c$.

In fact, $\mathbb{RP}^{4m+1}$ is a $\mathsf{spin}^{\mathbb{Z}_4}$ manifold. Not only that, we can uniformly construct a $\mathsf{spin}[k]$ structure on $\mathbb{RP}^n$, for $n + 1 + k \equiv 0 \mod 4$, as follows.

For this purpose, we first consider $\mathbb{R}^{n+1}$ and uplift the structure group of the (trivial) tangent bundle $T\mathbb{R}^{n+1}$ to $\mathrm{Spin}(n+1;k)$. Then we take an element $\mathsf{R} \in \mathrm{Spin}(n+1;k)$ on $\mathbb{R}^{n+1}$ given by

$$\mathsf{R} := (\Gamma^0 \Gamma^1 \cdots \Gamma^n)(\Gamma^{n+1} \cdots \Gamma^{n+k}). \tag{3.11}$$

When $n + 1 + k \equiv 0 \mod 4$, the $\mathsf{R}$ defined above satisfies the following two conditions:

$$\mathsf{R}^2 = 1, \qquad \mathsf{R}^\dagger \Gamma^I \mathsf{R} = -\Gamma^I. \tag{3.12}$$

For example, the first is satisfied since $\mathsf{R}^2 = (-1)^{\frac{1}{2}(n+k)(n+k+1)}$. Then we can define a manifold $\mathbb{RP}^n = S^n/\{1, \mathsf{R}\}$ equipped with the $\mathrm{Spin}(n;k)$ structure which follows naturally from the $\mathrm{Spin}(n+1;k)$ bundle on $\mathbb{R}^{n+1}$.

Note that this $\mathsf{R}$ is mapped to

$$\rho_1(\mathsf{R}) = -I_{n+1} \in \mathrm{O}(n+1), \qquad \rho_2(\mathsf{R}) = -1 \in \mathbb{Z}_2. \tag{3.13}$$

Because of the latter relation, we see that the generator $a \in H^1(\mathbb{RP}^n, \mathbb{Z}_2)$ is the holonomy of the $\mathbb{Z}_2$ bundle.

### 3.3  O4 and $\mathbb{RP}^4$

Let us first recall the case of O4-planes and D2-branes, which were discussed at length in [8] in the context of M2-branes in non-orientable spacetime in M-theory. Here the angular direction $\mathbb{RP}^4$ has the $\text{pin}^+ = \text{spin}[3]$ structure, and therefore we need to consider $\Omega_4^{\text{pin}^+}$. This group is known to be $\mathbb{Z}_{16}$ generated by $\mathbb{RP}^4$, and the $\eta$ invariant of a $\text{pin}^+$ Majorana fermion of $\mathbb{RP}^4$ is known to give $\exp(2\pi i/16)$ [36, 39]. This $\mathbb{Z}_{16}$ famously classifies the interacting fermionic SPTs in 3+1d [40–44], and also the anomaly of time-reversal invariant fermionic systems in 2+1d [11, 45], both with $\mathsf{T}^2 = (-1)^F$.

That the gaugino on the D2-brane is a representation of $\text{pin}^+$ can be seen as follows: the IIA theory itself has the $\text{pin}^+$ structure which naturally follows from the $\text{pin}^+$ structure of M-theory (see appendix A), and the supercharge preserved by D2-brane is $Q^{\text{D2}} = Q + \Gamma^9 \cdots \Gamma^3 \tilde{Q}$ (2.31). Therefore, a spatial reflection along $\Gamma^I$ ($I = 1, 2$) on the D2-brane worldvolume acts on the supercharge by $\Gamma^9 \cdots \Gamma^3 \Gamma^I$, which squares to $+1$.

The worldvolume theory of a D2-brane is $\mathcal{N} = 8$ supersymmetric. As such, the corresponding $\eta$ invariant in four dimensions is the eight times the generator, and assigns $-1$ to $\mathbb{RP}^4$. Since $\int_{\mathbb{RP}^4} w_4 = 1$, this $\eta$ invariant is given by $(-1)^{\int_M w_4}$ for general 4-dimensional $\text{pin}^+$ manifolds $M$.

### 3.4  O3 and $\mathbb{RP}^5$

In type IIB string theory, $\mathbb{RP}^5$ appears as the angular direction of an O3-plane. We already saw above that $\mathbb{RP}^5$ is orientable, not spin, but $\text{spin}^c$. That said, the natural structure in type IIB string theory is not quite the $\text{spin}^c$ structure. Rather, we are using the center $\mathbb{Z}_2 = \{1, \mathscr{C}\}$ of the $SL(2, \mathbb{Z})$ duality group. More precisely, what acts on the fermions is $\mathbb{Z}_4 = \{1, \mathscr{C}, \mathscr{C}^2, \mathscr{C}^3\}$, since $\mathscr{C}^2 = (-1)^F$ (see appendix A). Indeed, $\mathscr{C} = \Omega(-1)^F$ and this combination appears in (2.46) when $p = 3$. Then, the structure group $SO(d)$ is lifted not to $\text{Spin}(d)$ but to $(\text{Spin}(d) \times \mathbb{Z}_4)/\mathbb{Z}_2$ where the quotient identifies $(-1)^F \in \text{Spin}(d)$ and $\mathscr{C}^2 \in \mathbb{Z}_4$. This is an analogue of the $\text{spin}^c$ structure where we use $\mathbb{Z}_4$ instead of $U(1)$; for the lack of a better name, let us call this a $\text{spin}^{\mathbb{Z}_4}$ structure. Therefore what interests us in this case is $\Omega_5^{\text{spin}\mathbb{Z}_4}$. Notice that $\text{spin}^{\mathbb{Z}_4} = \text{spin}[2]$ in the notation of Sec. 3.1.

The basic feature of a $\text{spin}^{\mathbb{Z}_4} = \text{spin}[2]$ manifold $M_d$ is as follows. The natural map $\rho_2 : (\text{Spin}(d) \times \mathbb{Z}_4)/\mathbb{Z}_2 \to \mathbb{Z}_4/\mathbb{Z}_2 = \mathbb{Z}_2$ defines a class $a \in H^1(M_d, \mathbb{Z}_4/\mathbb{Z}_2)$, defining a real line bundle $L$ on $M_d$. The $\text{spin}^{\mathbb{Z}_4} = \text{spin}[2]$ structure on $M_d$ is a spin structure on $TM_d \oplus L \oplus L$. Therefore, we have the condition that $w_2(M_d) = a^2$. More physically, the bundle $L \oplus L$ may be regarded as the tangent bundle of the F-theory torus $T^2$ on which $\mathscr{C}$ acts as a 180° rotation as explained in appendix A.

The Poincaré dual of $a$ can be realized as a submanifold $N_{d-1} \subset M_d$, as we already saw in Sec. 3.1. Now we easily compute that

$$w_1(N_{d-1}) = a, \qquad w_2(N_{d-1}) = w_2(M_d) - a^2 = 0. \tag{3.14}$$

This means that $N_{d-1}$ has a $\text{pin}^+ = \text{spin}[3]$ structure. This construction is clearly cobordism invariant, and defines a homomorphism

$$s : \Omega_d^{\text{spin}\mathbb{Z}_4} \to \Omega_{d-1}^{\text{pin}+}. \tag{3.15}$$

This is the Smith homomorphism given in (3.8).

When $d = 5$, the Smith homomorphism clearly sends our $\mathbb{RP}^5$ to $\mathbb{RP}^4$, which we already recalled to be the generator of $\Omega_4^{\text{pin}+} = \mathbb{Z}_{16}$. In fact $\Omega_5^{\text{spin}\mathbb{Z}_4} \simeq \Omega_4^{\text{pin}+} = \mathbb{Z}_{16}$.[15]

---

[15]To see this, one identifies $\Omega_d^{\text{spin}\mathbb{Z}_4} \simeq \Omega_d^{\text{spin}}(B\mathbb{Z}_2, \xi)$ as explained in footnote 13. We then apply the Atiyah-

In terms of the anomaly of fermions, consider a 4d Majorana fermion $\psi$ on which $\mathbb{Z}_4$ acts as a multiplication by $i$. We can try to give a position-dependent mass term $m(x)\psi\psi$, but $m(x)$ is odd under $\mathbb{Z}_4/\mathbb{Z}_2$. Therefore, $m$ needs to be zero along the Poincaré dual $N_3$ of $a \in H^1(M_4, \mathbb{Z}_4/\mathbb{Z}_2)$. We now have a pin$^+$ fermion, i.e. a time-reversal-invariant fermion with $\mathsf{T}^2 = (-1)^F$ along $N_3$. This relates the anomaly of spin$^{\mathbb{Z}_4}$ fermions in 4d and that of pin$^+$ fermions in 3d, and the latter is famously characterized by $\mathbb{Z}_{16}$.

The worldvolume theory of a D3-brane is $\mathcal{N}{=}4$ supersymmetric. Therefore, it has four copies of this minimal amount of spin$^{\mathbb{Z}_4}$ fermion. The corresponding $\eta$ invariant in 5d then assigns four times the generator of $\mathbb{Z}_{16}$, i.e. $\exp(\pm\pi i/2)$, to $\mathbb{RP}^5$.

**Remark on S-fold:** The O3-plane has generalizations, called S-folds, which use more nontrivial elements of $SL(2,\mathbb{Z})$ than $\mathscr{C}$ [48,49]. The anomaly cancellation in this case is much more nontrivial than the O3-plane since the U(1) gauge field can contribute to the anomaly [50]. The relevant structure group is spin$^{\mathbb{Z}_{2k}} := (\text{spin} \times \mathbb{Z}_{2k})/\mathbb{Z}_2$, where $\mathbb{Z}_{2k}$ is a spin cover of the subgroup $\mathbb{Z}_k \subset SL(2,\mathbb{Z})$ for $k = 2,3,4,6$. (See appendix A for the reason why the spin cover of $SL(2,\mathbb{Z})$ appears.) It would be interesting to work out the details.

## 3.5 O2 and $\mathbb{RP}^6$

Here $\mathbb{RP}^6$ is pin$^-$ = spin[1], and therefore we are interested in $\Omega_6^{\text{pin}^-}$. In type IIA string theory, pin$^-$ is realized as follows. Recall that M-theory has pin$^+$. Then, in type IIA, whenever we act a spatial reflection in the direction $I \leq 9$, we combine it also with the reflection along the M-theory circle which we represent by $I = 10$. Because $(\Gamma^I\Gamma^{10})^2 = -1$ for $I \leq 9$, it becomes pin$^-$. The line bundle $L$ associated to the $\mathbb{Z}_2$ (orientation) bundle is physically interpreted as the tangent bundle of the M-theory circle.

The reflection along the M-theory circle is realized by $(-1)^{F_L}$ in the IIA theory, so we are combining the pin$^+$ reflections and $(-1)^{F_L}$ to realize pin$^-$. The factor $(-1)^{F_L}$ is seen in (2.46) when $p = 2$.

It is known that $\Omega_6^{\text{pin}^-} \simeq \mathbb{Z}_{16}$. The $\eta$ invariant of the minimal pin$^-$ fermion on $\mathbb{RP}^6$ is known to be $\exp(\pm\pi i/8)$, and therefore $\mathbb{RP}^6$ generates $\Omega_6^{\text{pin}^-}$.

That the gaugino on the D4-brane is a representation of pin$^-$ can be seen just as we considered the pin$^+$-ness of the D2-brane above. Namely, the supercharge preserved by D4-brane is $Q^{\text{D4}} = Q + \Gamma^9 \cdots \Gamma^5 \tilde{Q}$ (2.31). Therefore, a spatial reflection along $\Gamma^I$ ($I \leq 4$) on the D4-brane worldvolume acts on the supercharge by $\Gamma^9 \cdots \Gamma^5 \Gamma^I$, which squares to $-1$. The worldvolume theory on a D4-brane is $\mathcal{N}{=}2$ supersymmetric. Therefore, it has two copies of minimal fermions, and assigns $\exp(\pm\pi i/4)$ to $\mathbb{RP}^6$.

## 3.6 O1 and $\mathbb{RP}^7$

In this case, $\mathbb{RP}^7$ is spin, but $\Omega_7^{\text{spin}} = 0$. Therefore, we should be seeing something else.

Recall that in 6d, a chiral fermion has a perturbative gravitational anomaly. Correspondingly, the $\eta$ invariant of the smallest fermion in 7d is not a cobordism invariant. Instead, let us consider a pair of a Weyl fermion with positive chirality and a Weyl fermion with negative chirality. We can further introduce a $\mathbb{Z}_2$ symmetry and declare that the positive-chirality fermion is odd while the negative-chirality fermion is even. The perturbative gravitational anomaly is canceled, and the corresponding $\eta$ invariant in 7d gives a cobordism invariant for $\Omega_7^{\text{spin}}(B\mathbb{Z}_2)$.

---

Hirzebruch spectral sequence (AHSS) for the twisted spin bordisms, constructed in [46,47]. The $E^2$ page is in fact the same as the AHSS for $\Omega^{\text{spin}}(B\mathbb{Z}_2)$, and the difference is only in the differentials. From this one easily sees that $|\Omega_5^{\text{spin}^{\mathbb{Z}_4}}| \leq 16$. As we already saw that the homomorphism sends $\mathbb{RP}^5$ to $\mathbb{RP}^4$ and hence $|\Omega_5^{\text{spin}^{\mathbb{Z}_4}}| \geq 16$, we are done.

The $\mathbb{Z}_2$ symmetry relevant for our string theory setup is generated by $\Omega$ as can be seen in (2.46) with $p = 1$. Therefore the relevant structure is $\mathsf{spin} \times \mathbb{Z}_2 = \mathsf{spin}[4]$.

As discussed in Sec. 3.1, we can define a homomorphism

$$s : \Omega_d^{\mathsf{spin}}(B\mathbb{Z}_2) \to \Omega_{d-1}^{\mathsf{pin}-}. \tag{3.16}$$

This was defined as follows: an element of $\Omega_d^{\mathsf{spin}}(B\mathbb{Z}_2)$ comes from a pair of a spin manifold $M_d$ and an element $a \in H^1(M_d, \mathbb{Z}_2)$. The Poincaré dual of $a$ can be represented by a submanifold $N_{d-1} \subset M_d$, and $N_{d-1}$ is $\mathsf{pin}^-$. The kernel of $s$ clearly includes $\Omega_d^{\mathsf{spin}} \subset \Omega_d^{\mathsf{spin}}(B\mathbb{Z}_2)$, and therefore we have a homomorphism

$$s : \tilde{\Omega}_d^{\mathsf{spin}}(B\mathbb{Z}_2) \to \Omega_{d-1}^{\mathsf{pin}-}, \tag{3.17}$$

where $\tilde{\Omega}_d^{\mathsf{spin}}(B\mathbb{Z}_2)$ is the reduced bordism group $\Omega_d^{\mathsf{spin}}(B\mathbb{Z}_2) = \Omega_d^{\mathsf{spin}} \oplus \tilde{\Omega}_d^{\mathsf{spin}}(B\mathbb{Z}_2)$.

A homomorphism in the inverse direction $t : \Omega_{d-1}^{\mathsf{pin}-} \to \tilde{\Omega}_d^{\mathsf{spin}}(B\mathbb{Z}_2)$ can be constructed as follows. We consider the $\mathbb{R}^2$-bundle $L \oplus \mathbb{R}$ over $N_{d-1}$, where $L$ is the orientation line bundle and $\mathbb{R}$ is a trivial real line bundle. We let $M_d'$ be the unit circle bundle over $N_{d-1}$ constructed from this $L \oplus \mathbb{R}$. This $M_d'$ has a natural spin structure, since the $\mathsf{pin}^-$ structure on $N_{d-1}$ gives the spin structure on $TN_{d-1} \oplus L$, which gives the spin structure on the total space of the bundle $L \oplus \mathbb{R}$ over $N_{d-1}$, and the $M_d'$ is the boundary of the unit disk bundle $D \subset L \oplus \mathbb{R}$ over $N_{d-1}$. However, it is bordant to the empty manifold since $\partial D = M_d'$. Now we introduce an additional $\mathbb{Z}_2$ holonomy $a \in H^1(M_d, \mathbb{Z}_2)$ which assigns $-1$ to the circle fiber. We denote this spin manifold equipped with the $\mathbb{Z}_2$ holonomy as $M_d$. This construction gives a map

$$t : \Omega_{d-1}^{\mathsf{pin}-} \to \tilde{\Omega}_d^{\mathsf{spin}}(B\mathbb{Z}_2). \tag{3.18}$$

Notice that the codomain of this map can be taken to be the subgroup $\tilde{\Omega}_d^{\mathsf{spin}}(B\mathbb{Z}_2) \subset \Omega_d^{\mathsf{spin}}(B\mathbb{Z}_2)$, because if we neglect the $\mathbb{Z}_2$ holonomy, it is bordant to $\varnothing$ by the disk bundle as discussed above. The two homomorphisms $s$ and $t$ satisfy $s \circ t = \mathrm{id}$. Therefore we see that $\Omega_{d-1}^{\mathsf{pin}-}$ is a direct summand of $\tilde{\Omega}_d^{\mathsf{spin}}(B\mathbb{Z}_2)$. In fact, the homomorphism (3.17) is an isomorphism for general $d$, known as the Smith isomorphism.[16]

We already saw that $\Omega_6^{\mathsf{pin}-} = \mathbb{Z}_{16}$ is generated by $\mathbb{RP}^6$. Combining the Smith isomorphism and $\Omega_7^{\mathsf{spin}} = 0$, we see that $\Omega_7^{\mathsf{spin}}(B\mathbb{Z}_2) = \mathbb{Z}_{16}$ generated by $\mathbb{RP}^7$ with a $\mathbb{Z}_2$ bundle whose $w_1$ is the generator of $H^1(\mathbb{RP}^7, \mathbb{Z}_2)$.

The $\eta$ invariant of a minimal 7d fermion on $\mathbb{RP}^7$, with a standard choice of the metric, is $\pm 1/32$, where the sign depends on the choice of the spin structure. The $\eta$ invariant of a minimal 7d fermion depends continuously on the metric, however; a cobordism invariant is given by taking the difference, which assigns $\exp(\pm \pi i / 8)$ to $\mathbb{RP}^7$.

The worldvolume theory on a D5-brane is $\mathcal{N}=(1,1)$ supersymmetric. Therefore, the perturbative gravitational anomaly cancels, and there is a $\mathbb{Z}_2$ symmetry $\Omega$ such that the mass term for the fermions is odd under it. This $\mathbb{Z}_2$ symmetry generated by $\Omega$ is a part of the duality symmetry of type IIB superstring, under which $g$, $C_2$ are even while $C_0$, $B_2$ and $C_4$ are odd. The $\mathbb{RP}^7$ around an O1 has a nontrivial holonomy of this $\mathbb{Z}_2$ symmetry as shown in (2.46). Therefore, the $\eta$ invariant controlling the global anomaly of the worldvolume fermions on a D5-brane is $\exp(\pm \pi i / 8)$.

---

[16] A proof goes as follows. We need to show that the kernel of $s$ is trivial. So, let us assume $N^{(d-1)}$ is zero in $\Omega_{d-1}^{\mathsf{pin}-}$. Then there is a $\mathsf{pin}^-$ manifold $W^{(d)}$ such that $\partial W^{(d)} = N^{(d-1)}$. Next we consider the orientation line bundle $Z^{(d+1)}$ over $W^{(d)}$. $Z^{(d+1)}$ is spin and has a canonical $\mathbb{Z}_2$ bundle on it. $Z^{(d+1)}$ is a manifold with corners. One boundary is of the form $\mathbb{R} \times N^{(d-1)}$, which we paste to the tubular neighborhood of $N^{(d-1)}$ in $M^{(d)} \times \{1\}$ in $M^{(d)} \times [0, 1]$. Then $M^{(d)} \times [0, 1] \sqcup Z^{(d+1)}$ provides a bordism from $M^{(d)}$ to a certain $X^{(d)}$. Now $X^{(d)}$ is spin, and the $\mathbb{Z}_2$ bundle on it is trivial since the Poincaré dual of the $\mathbb{Z}_2$ holonomy can be taken to be given by $N^{(d-1)} \times [0, 1] \sqcup W^{(d)}$ which does not touch $X^{(d)}$. Therefore it is in $\Omega_d^{\mathsf{spin}}$, q.e.d. The authors thank R. Thorngren for providing this proof.

**GSO projection in string theory:**    In passing, we mention that the same argument shows that $\Omega_3^{\mathrm{spin}}(B\mathbb{Z}_2) = \Omega_2^{\mathrm{pin}-} = \mathbb{Z}_8$. On the one hand, $\Omega_2^{\mathrm{pin}-} = \mathbb{Z}_8$ is generated by $\mathbb{RP}^2$, whose Brown-Arf invariant is $1/8$. On the other hand, $\Omega_3^{\mathrm{spin}}(B\mathbb{Z}_2) = \mathbb{Z}_8$ is generated by $\mathbb{RP}^3$, and classifies the anomaly of 2d systems with equal numbers of left-moving and right-moving fermions, with an additional $\mathbb{Z}_2$ symmetry. The corresponding $\mathbb{Z}_2$ bundle can be thought of as measuring the difference of the left-moving spin structure and the right-moving spin structure.

Consider $N_f$ non-chiral 2d Majorana fermions. The $\eta$ invariant assigns $\exp(\pm N_f \pi i/4)$ to $\mathbb{RP}^3$. Therefore, only when $N_f$ is a multiple of 8, the system is non-anomalous and we can perform the sum over the left-moving and the right-moving spin structures independently, or equivalently perform the chiral Gliozzi-Scherk-Olive projection [51]. Luckily, the superstring worldsheet in the light-cone gauge has $N_f = 8$. This is another mathematical accident underlying the consistency of superstring theory. This anomaly cancellation for $N_f = 8$ was first shown in [52].

## 3.7  O0 and $\mathbb{RP}^8$

Finally we briefly mention the case of O0 and $\mathbb{RP}^8$. Here $\mathbb{RP}^8$ is $\mathrm{pin}^+$. This leads us to examine $\Omega_8^{\mathrm{pin}+}$. This is known to be $\simeq \mathbb{Z}_2 \times \mathbb{Z}_{32}$. The $\eta$ invariant of the minimal $\mathrm{pin}^+$ fermion on $\mathbb{RP}^8$ is known to be $\exp(\pm \pi i/16)$, and is the generator of the $\mathbb{Z}_{32}$ part.

The worldvolume theory on a D6-brane is $\mathcal{N}{=}1$ supersymmetric. Therefore, it assigns to $\mathbb{RP}^8$ the phase $\exp(\pm \pi i/16)$.

## 3.8  Summary

We can summarize the discussions in this section as follows:

| orientifold | O4 | O3 | O2 | O1 | O0 |
|---|---|---|---|---|---|
| projective space | $\mathbb{RP}^4$ | $\mathbb{RP}^5$ | $\mathbb{RP}^6$ | $\mathbb{RP}^7$ | $\mathbb{RP}^8$ |
| cobordism group | $\Omega_4^{\mathrm{pin}+}$ | $\Omega_5^{\mathrm{spin}\mathbb{Z}_4}$ | $\Omega_6^{\mathrm{pin}-}$ | $\Omega_7^{\mathrm{spin}}(B\mathbb{Z}_2)$ | $\Omega_8^{\mathrm{pin}+}$ |
| · in our notation | $\Omega_4^{\mathrm{spin}[3]}$ | $\Omega_5^{\mathrm{spin}[2]}$ | $\Omega_6^{\mathrm{spin}[1]}$ | $\Omega_7^{\mathrm{spin}[4]}$ | $\Omega_8^{\mathrm{spin}[3]}$ |
| · explicitly | $\mathbb{Z}_{16}$ | $\mathbb{Z}_{16}$ | $\mathbb{Z}_{16}$ | $\mathbb{Z}_{16}$ | $\mathbb{Z}_{32} \times \mathbb{Z}_2$ |
| D-brane involved | D2 | D3 | D4 | D5 | D6 |
| susy on the D-brane | $\mathcal{N}{=}8$ | $\mathcal{N}{=}4$ | $\mathcal{N}{=}2$ | $\mathcal{N}{=}(1,1)$ | $\mathcal{N}{=}1$ |
| $\pm$orientifold charge | $8 \times \frac{1}{16} = \frac{1}{2}$ | $4 \times \frac{1}{16} = \frac{1}{4}$ | $2 \times \frac{1}{16} = \frac{1}{8}$ | $1 \times \frac{1}{16} = \frac{1}{16}$ | $1 \times \frac{1}{32} = \frac{1}{32}$ |

.

# Acknowledgements

The authors would like to thank Y. Imamura for correspondence on supergravity. Y.T. is partially supported by JSPS KAKENHI Grant-in-Aid (Wakate-A), No.17H04837 and JSPS KAKENHI Grant-in-Aid (Kiban-S), No.16H06335, and also by WPI Initiative, MEXT, Japan at IPMU, the University of Tokyo.  K.Y. is supported by JSPS KAKENHI Grant-in-Aid (Wakate-B), No.17K14265.

# A   The duality group of type IIB string theory

It is often said in the literature that the duality group of type IIB string theory is $\mathrm{SL}(2,\mathbb{Z})$. This is not entirely precise: as we saw, the O3 background uses $\mathscr{C} \in \mathrm{SL}(2,\mathbb{Z})$, but more precisely $\mathscr{C}^2 = (-1)^F$. We also saw that the O1 background uses the $\mathbb{Z}_2$ symmetry $\Omega$, which is not even a part of $\mathrm{SL}(2,\mathbb{Z})$ but has nontrivial commutation relations with elements of $\mathrm{SL}(2,\mathbb{Z})$.

In this short appendix we specify the duality group more precisely. The basic statement is as follows:[17]

> *The duality group of type IIB string theory is the* pin$^+$ *version of the double cover of* GL(2, $\mathbb{Z}$).

Let us first recall the massless fields in type IIB supergravity. The bosons are the dilaton $e^{-\phi}$, the metric $g$, the NS-NS B-field $B_2$, and the R-R fields $C_0$, $C_2$, $C_4$; the fermions consist of the dilatinos $\psi$, and the gravitinos $\Psi_I$.

**Non-perturbative symmetries:** The action of the non-perturbative symmetry SL(2, $\mathbb{Z}$) on the bosonic fields is well-known. Schematically, we can say that $g$ and $C_4$ are invariant; two two-forms $(B_2, C_2)$ transform as a doublet; and the combination $\tau := ie^{-\phi} + C_0$ is acted on by the fractional linear transformation. The center of SL(2, $\mathbb{Z}$) is $\mathbb{Z}_2 = \{1, \mathscr{C}\}$. When we consider the action on fermions, it is known that $\mathscr{C}$ in fact squares to $(-1)^F$ [55, 56]. This gives a nontrivial extension of SL(2, $\mathbb{Z}$) by $\mathbb{Z}_2 = \{1, (-1)^F\}$ known as Mp(2, $\mathbb{Z}$).

**Perturbative symmetries:** From the perturbative worldsheet perspective, we see two $\mathbb{Z}_2$ symmetries, often denoted as $\Omega$ and $(-1)^{F_L}$. The former comes from the worldsheet orientation reversal, and the latter is the left-moving fermion number. The parity under these two $\mathbb{Z}_2$ operations of bosonic fields are as follows:

$$
\begin{array}{c||cccc|cc}
 & \phi & g & C_0 & C_4 & B_2 & C_2 \\
\hline
\mathscr{R}_1 := \Omega & + & + & - & - & - & + \\
\mathscr{R}_2 := (-1)^{F_L} & + & + & - & - & + & - \\
\end{array}
\tag{A.1}
$$

Here, we introduced another notations $\mathscr{R}_{1,2}$ for $\Omega$ and $(-1)^{F_L}$. This is because $\mathscr{R}_{1,2}$ extends SL(2, $\mathbb{Z}$) to GL(2, $\mathbb{Z}$): $\mathscr{R}_a$ corresponds to the flip of the $a$-th coordinate, and sends $\tau \to -\bar{\tau}$. We also see that $\mathscr{R}_1 \mathscr{R}_2 = \mathscr{C}$, as it should be within GL(2, $\mathbb{Z}$).

Now let us discuss the actions on the fermions. The worldsheet computation shows that

$$
\mathscr{R}_1{}^2 = \mathscr{R}_2{}^2 = 1.
\tag{A.2}
$$

But they do not commute [3, Sec. 2.2]: Indeed, from the worldsheet point of view, we have

$$
\Omega(-1)^{F_L} = (-1)^{F_R}\Omega = (-1)^F(-1)^{F_L}\Omega,
\tag{A.3}
$$

which in terms of $\mathscr{R}_{1,2}$ is given by

$$
\mathscr{R}_1 \mathscr{R}_2 = (-1)^F \mathscr{R}_2 \mathscr{R}_1.
\tag{A.4}
$$

As an example, the action of them on the two supercharges $Q$ and $\tilde{Q}$ of type IIB string theory discussed in Sec. 2.2 is given as

$$
\mathscr{R}_1 \begin{pmatrix} Q \\ \tilde{Q} \end{pmatrix} = \begin{pmatrix} 0 & 1 \\ 1 & 0 \end{pmatrix} \begin{pmatrix} Q \\ \tilde{Q} \end{pmatrix}, \qquad \mathscr{R}_2 \begin{pmatrix} Q \\ \tilde{Q} \end{pmatrix} = \begin{pmatrix} 1 & 0 \\ 0 & -1 \end{pmatrix} \begin{pmatrix} Q \\ \tilde{Q} \end{pmatrix}.
\tag{A.5}
$$

The relations (A.2) and (A.4) modulo $(-1)^F$ defines a $\mathbb{Z}_2 \times \mathbb{Z}_2$ subgroup of O(2). The element $(-1)^F$ then extends O(2) to Pin$^+$(2); $\mathscr{R}_1$ and $\mathscr{R}_2$ generate a subgroup isomorphic to the dihedral group with eight elements [3, Sec. 2.2].

Combining the observations above, we see that, when the action on the fermions are taken into account, GL(2, $\mathbb{Z}$) is extended to a double cover of pin$^+$ type.

---

[17] It would be interesting to study the cancellation of the anomaly of this duality group in type IIB theory. Previous analyses can be found e.g. in [53, 54].

**F-theory interpretation:** The structure we saw above has a natural F-theory interpretation. We realize the F-theory as a limit of M-theory on a spacetime which is a $T^2$-fibration. As is well-known, the SL(2, $\mathbb{Z}$) duality acts on $T^2$ as a change of basis of determinant one, and the combination $\tau = ie^{-\phi} + C_0$ is the complex modulus of $T^2$.

To see why SL(2, $\mathbb{Z}$) is extended to Mp(2, $\mathbb{Z}$), recall that the central element $\mathscr{C} \in$ SL(2, $\mathbb{Z}$) corresponds to a 180° rotation of the $T^2$ fiber. Clearly, $\mathscr{C}^2$ then corresponds to a 360° rotation, which acts by $(-1)^F$. This means that $\mathscr{C}$ is in fact of order four, extending SL(2, $\mathbb{Z}$) to Mp(2, $\mathbb{Z}$).

To see why SL(2, $\mathbb{Z}$) is extended to GL(2, $\mathbb{Z}$), we realize that M-theory is in fact parity symmetric. Therefore, we can consider the action of flipping only one coordinate of $T^2$. When we take the F-theory limit, this becomes a purely internal operation, which does not act on the spacetime of type IIB theory. They provide the operations $\mathscr{R}_1$ and $\mathscr{R}_2$.

Finally, to see why the double cover of GL(2, $\mathbb{Z}$) should be of type pin$^+$ rather than pin$^-$, we recall that M-theory itself is pin$^+$ rather than pin$^-$. This is simply because the eleven-dimensional supercharge is a Majorana fermion, which is a representation of pin$^+$ but not of pin$^-$.

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
