# Peer review of "Why are fractional charges of orientifolds compatible with Dirac quantization?"

_SciPost Physics, doi:SciPost Phys. 7, 058 (2019)_

## Round 1 · Referee Report · Anonymous (Referee 1) · 2018-6-16

Strengths
The paper resolved a long-standing issue of the fractional RR-charges carried by some lower dimensional Orientifold planes, which seemingly contradict the Dirac quantization. The resolution is both elegant and instructive.
Weaknesses
None.
Report
Although vanilla D-brane charges are well-known to obey and saturate the Dirac quantization rule, there are objects in superstring theories that carry fractional charges. The most prominent such are Orientifold planes in lower dimensions; the problematic Aharanov-Bohm phase acquired by the dual Dq-brane circling around such an Op plane has long been unexplained.
This paper offers a clean resolution by demonstrating how this phase gets canceled precisely by an adiabatic phase acquired by the worldvolume fermion determinant on the world-volume. The latter phase is computed by the eta invariant on the (q+2)-dimensional subspace swept by Dq-brane wolrd-volume.
The authors in part borrow from existing math computation of the eta for real projective spaces, and in part relies on a very elegant and universal way to evaluate them. The latter is done by observing that the counting of zero modes on $T^(q+3)$ even under $Z_2$ projection should equal the APS index on the same $T^(q+3)/Z_2$ with the conical singularities removed. Section 2 includes a careful treatment of the $q$-dependent sign of the eta, which is also crucial for the purported cancelation.
Overall, the paper is very impressive.
This paper offers a clean resolution by demonstrating how this phase gets canceled precisely by an adiabatic phase acquired by the worldvolume fermion determinant on the world-volume. The latter phase is computed by the eta invariant on the (q+2)-dimensional subspace swept by Dq-brane wolrd-volume.
The authors in part borrow from existing math computation of the eta for real projective spaces, and in part relies on a very elegant and universal way to evaluate them. The latter is done by observing that the counting of zero modes on $T^(q+3)$ even under $Z_2$ projection should equal the APS index on the same $T^(q+3)/Z_2$ with the conical singularities removed. Section 2 includes a careful treatment of the $q$-dependent sign of the eta, which is also crucial for the purported cancelation.
Overall, the paper is very impressive.
Requested changes
The only place that could be perhaps improved for clarity is the top half of page 5, a review of Gilkey's result, which I find to be a little too sketchy to be useful.

---

## Round 1 · Referee Report · Anonymous (Referee 2) · 2018-7-1

Strengths
- Provides a rigorous understanding of the geometry around generic orientifold planes and its impact on flux quantisation, generalising previous results.
- Makes, and justifies, a precise claim about the duality group of the type IIB string.
Weaknesses
- The paper is admittedly rather technical and would be of interest primarily to those interested in mathematical subtleties. At the same time, it should help avoid confusions (including sign confusions) when future authors use orientifolds in physical situations. Also its relation to SPT phases will be of interest in a different community.
Report
In this manuscript, the authors examine consistency of the fractional charge on orbifolds with the Dirac quantisation condition which naively suggests that all such charges should be integral. In the case of O4 planes, they recall old observations by Witten who showed that the 4-form flux in M/type IIA theory is related to half the Stiefel-Whitney class of the ambient space. The latter object is a half-integer in the presence of an O4-plane. Indeed the above result can be interpreted in terms of the eta-invariant of the “sphere” surrounding the plane, which due to the $Z_2$ involution defining the ambient space, is really Rp$^4$. Since the eta invariant of this space is half-integral, the 4-form flux must be half-integral as well.
For Op planes with $p<4$, the goal then is to show that the eta-invariant for Rp$^{8-p}$ is fractionally quantised in positive or negative integer multiples of $2^{p-5}$. The authors show this using the equivariant version of Atiyah-Patodi-Singer. The basic result is presented in Section 1, Section 2 is devoted to spelling out the details (including the signs, which are important). In Section 3 the authors go through all the cases individually and identify relations to SPT phases.
Finally in an Appendix they argue that the duality group of type IIB string theory is not SL(2,Z) but rather the pin$^+$ double cover of GL(2,Z).
For Op planes with $p<4$, the goal then is to show that the eta-invariant for Rp$^{8-p}$ is fractionally quantised in positive or negative integer multiples of $2^{p-5}$. The authors show this using the equivariant version of Atiyah-Patodi-Singer. The basic result is presented in Section 1, Section 2 is devoted to spelling out the details (including the signs, which are important). In Section 3 the authors go through all the cases individually and identify relations to SPT phases.
Finally in an Appendix they argue that the duality group of type IIB string theory is not SL(2,Z) but rather the pin$^+$ double cover of GL(2,Z).
Requested changes
- I suspect that some of the observations in the Appendix could be (at least implicitly) contained in the works of Sen, hep-th/9603113 and hep-th/9604070, as well as subsequent works hep-th/9810153 and hep-th/9810213 by other authors. It would be nice if the authors of the present work could identify/clarify some of the observations in those papers in terms of their elegant and mathematically rigorous formalism. If the observations in those papers turn out to be unrelated to the present work, then too it would be nice to say so.

---

## Round 2 · Author Response

We are sorry for taking a very long time to revise and resubmit the version 2. This was due to a sign error we noticed around when we recevied the referee reports, which tooks us many months to resolve. In this paper we decided to stick to the O+ planes. The O- planes will be treated in a separate future publication; the anlaysis of the O3- plane was already given in our paper https://arxiv.org/abs/1905.08943 (with Chang-Tse Hsieh) .

---

## Round 2 · List of Changes

- We clarified that we only treat O+ planes. O- planes also induce anomalies of the Maxwell theory on the D-brane worldvolume around them, and are much harder to study.

- Referee 1 suggested that we might improve our brief summary of Gilkey's method to compute the eta invariant. We opted not to follow this advice, as this topic would be covered in more detail in our forthcoming paper (with Chang-Tse Hsieh) on the issue of fractional charges of both O+ and O- planes.

- Referee 2 suggested us to look at four papers related to our Appendix.

As for two papers by Sen hep-th/9603113 and hep-th/9604070, Sen indeed discussed a somewhat related issue, but not exactly on how the SL(2,Z) of IIB needs to be interpreted. There were also some other papers from that period discussing issues similar to what Sen discussed. But they are all cited in Dabholkar's review hep-th/9804208, which we did cite and was actually the input of our discussion in the Appendix, as can be seen above our eq. (A.3). So we consider Sen's two papers already implicitly cited.

As for two papers hep-th/9810153 and hep-th/9810213 on SL(2,Z) anomaly of type IIB theory, we agree that this is an interesting question. But our objective in the Appendix was to clarify the precise group structure of the duality group, and studying its anomaly is something beyond our aim in this paper. We added a brief foonote concening this point.

- We also added other minor clarifications.

You are currently on this page

Resubmission 1805.02772v2 on 4 October 2019

---

## Editorial Decision

published